



# Using spectra characteristics to identify ice nucleating particle
# populations during winter storms in the Alps
Jessie M. Creamean[1,2*], Claudia Mignani[3], Nicolas Bukowiecki[4], Franz Conen[3]
[1]Cooperative Institute for Research in Environmental Sciences, University of Colorado, Boulder, CO, USA
[2]Physical Sciences Division, National Oceanic and Atmospheric Administration, Boulder, CO, USA
[3]Department of Environmental Sciences, University of Basel, Switzerland
[4]Laboratory of Atmospheric Chemistry, Paul Scherrer Institute, Villigen, Switzerland
[*]Now at: Department of Atmospheric Science, Colorado State University, Fort Collins, CO, USA
Email: jessie.creamean@colostate.edu
**Abstract.** One of the least understood cloud processes is modulation of their microphysics by aerosols, specifically of cloud ice
by ice nucleating particles (INPs). To investigate INP impacts on cloud ice and subsequent precipitation formation, measurements
in cloud environments are necessary but difficult given the logistical challenges associated with airborne measurements and
separating interstitial aerosol from cloud residues. Additionally, determining the sources of INPs is important given the dependency
of glaciation temperatures on the mineral or biological components and diversity of such INP populations. Here, we present results
from a comparison of INP spectral characteristics in air, cloud rime, and fresh fallen snow for storm days at the High-Altitude
Research Station, Jungfraujoch. The goal of the study was two-fold: (1) to assess variability in wintertime INP populations found
in-cloud based on air mass direction during snowfall and (2) to evaluate INPs between different sample types using normalized
differential INP spectra. INP freezing temperatures and concentrations were consistently higher on average from the southeast as
compared to the northwest for rime, snow and especially aerosol samples which is likely a result of air mass influence from
boundary layer terrestrial and marine sources in Southern Europe, the Mediterranean, and North Africa. For all three sample types
combined, average onset freezing temperatures were –7.7 and –12 °C for southeasterly and northwesterly days, respectively, while
INP concentrations were 3 to 20 times higher when winds arrived from the southeast. Southeasterly aerosol samples typically had
bimodal freezing spectra—indicating a putative influence from biological sources—while bimodality of the rime and snow varied
depending on meteorological context. Evaluating normalized differential INP spectra exhibited variable modality and shape—
depending on the types of INPs present—and may serve as a viable method for comparing different sampling substances and
assessing the possible mixed mineral and biological versus only biological contributions to INP sample populations.
## 1 Introduction
Aerosols are key players in the atmospheric radiation budget, cloud microphysics, and precipitation development. However, one
of the most significant challenges with regard to aerosols is quantifying their impacts on cloud ice formation through serving as
ice nucleating particles (INPs) (Boucher et al., 2013). Aerosol-induced ice microphysical modifications influence cloud lifetime
and albedo (Albrecht, 1989; Twomey, 1977; Storelvmo et al., 2011), as well as the production of precipitation in clouds containing
both liquid and ice. Mixed-phase clouds (MPCs) are ubiquitous in the troposphere over the entire annual cycle yet are difficult to
quantify globally in part due to an inadequate understanding of aerosol-cloud interactions in mixed-phase environments (Korolev
et al., 2017). Thus, a close evaluation of aerosol-cloud processes is crucial to evaluating weather and climate processes; however,
constraining aerosol-cloud impacts in models, specifically when parameterizing INPs in MPC systems, remains a significant
challenge due to limited observations (Cziczo et al., 2017; Coluzza et al., 2017; DeMott et al., 2010; Kanji et al., 2017; Korolev et
al., 2017). Observations directly in cloudy environments are even more scarce—given the logistical costs and resources required



by airborne platforms, caveats associated with aircraft probes and instrumentation, and instrumental artefacts caused by flying
through clouds at high speeds (Cziczo et al., 2017)—but are necessary to assess the impacts of INPs on MPC microphysics as
compared to most surface measurements which are geared towards evaluation of INP sources.
In the absence of conditions with −38 °C and relative humidity with respect to ice above 140%, INPs are required for initiation of
tropospheric cloud ice formation (Kanji et al., 2017). Aerosols such as dust and primary biological aerosol particles (PBAPs) are
some of the most abundant and efficient INPs found in the atmosphere, respectively (Murray et al., 2012; Hoose and Möhler, 2012;
DeMott et al., 1999; Conen et al., 2011; Creamean et al., 2013). PBAPs originating from certain bacteria, pollens, and vegetative
detritus are the most efficient INPs known, capable of initiating freezing near −1 °C, while most PBAPs (e.g., fungal spores, algae,
and diatoms) tend to nucleate ice at temperatures similar to those of mineral dust (Despres et al., 2012; Murray et al., 2012; Tobo
et al., 2014; Hader et al., 2014a; O'Sullivan et al., 2014; Hill et al., 2016; Tesson et al., 2016; Alpert et al., 2011; Knopf et al., 2010;
Fröhlich-Nowoisky et al., 2015). In general, previous works collectively indicate that PBAP INPs that nucleate ice greater than
approximately –10 °C are bacterial (Murray et al., 2012; Hu et al., 2018; Hoose and Möhler, 2012; Despres et al., 2012; Frohlich-
Nowoisky et al., 2016), but could also be pollen or certain fungal spores (von Blohn et al., 2005; Hoose and Möhler, 2012;
O'Sullivan et al., 2016), although the latter two are less likely. Plant bacteria such as *Pseudomonas syringae* are deemed
omnipresent in the atmosphere and precipitation (Despres et al., 2012; Stopelli et al., 2017; Morris et al., 2014), and facilitate cloud
ice formation up to −1 °C (Despres et al., 2012). While, only a few laboratory-based studies have reported known inorganic or
mineral materials that ice nucleation activity at such temperatures (Ganguly et al., 2018; Atkinson et al., 2013). Mineral and soil
dust serving as atmospheric shuttles for organic microbial fragments can be transported thousands of kilometres and serve as
effective INPs, even from highly arid regions such as the Sahara (Kellogg and Griffin, 2006), yet the exact origin of the ice
nucleation germ forming at the warmest temperatures is speculated to be due to the ice binding proteins or macromolecules of the
biological components (O'Sullivan et al., 2014; O'Sullivan et al., 2016; Conen and Yakutin, 2018). In general, the previous studies
on the climate relevance of PBAPs demonstrate the importance of such INPs at MPC temperatures and precipitation enhancement
(Morris et al., 2004; Bergeron, 1935; Christner et al., 2008; Morris et al., 2014; Morris et al., 2017; Stopelli et al., 2014; Frohlich-
Nowoisky et al., 2016).
Although biological constituents, from cellular material to in-tact bacteria and spores, are thought to be omnipresent in the
atmosphere (Burrows et al., 2009b; Burrows et al., 2009a; Jaenicke, 2005; Jaenicke et al., 2007), modeling studies constraining
global emission estimates of biological INPs and PBAPs are very limited, subject to significant hurdles, and often yield conflicting
results due to the dearth of observations and complexity of atmospheric PBAPs (Hummel et al., 2015; Burrows et al., 2013; Twohy
et al., 2016; Frohlich-Nowoisky et al., 2012; Despres et al., 2012; Hoose and Möhler, 2012; Morris et al., 2011). Yet, biological
aerosols such as bacteria have been shown to cause significant perturbations in cloud ice in numerical weather prediction models,
affording modulations in cloud radiative forcing and precipitation formation (Sahyoun et al., 2017). In addition, measuring and
quantifying PBAPs is non-trivial—methodologies for counting, culturing, and nucleic acid sequencing of PBAPs and especially
for those which fall in the warm temperature INP regime (i.e., INPs that nucleate ice > −15 °C) are: (1) time and labor intensive,
(2) require specific expertise or at times substantial resources, (3) require substantial sample volumes, or (4) are species- or genera-
specific or limited to viable microorganisms (Despres et al., 2012). Although such techniques are required to adequately assess the
atmospheric microbiome and PBAP sources, a simpler approach could be applied to evaluate and even quantify warm temperature
biological INP populations as compared to colder temperature PBAPs or mineral dust.





The goal of the study presented here focuses on: (1) an intercomparison of INP measurements of aerosol, cloud rime, and snow directly in cloudy environments at the ground and (2) evaluating INP spectra in a manner such that we can estimate the relative contribution from biological INPs in the warm temperature regime relevant to MPCs. Sampling was conducted at the High-Altitude Research Station Jungfraujoch (JFJ), a unique location for evaluating populations of INPs that affect winter storms in the European Alps, and where MPCs are particularly common (Lohmann et al., 2016). Recent studies at JFJ have provided valuable insight into INP concentrations, sources, and removal processes under a variety of conditions and during various times of the year. Conen et al. (2015) measured INPs at –8 ˚C over the course of a year at JFJ, and found a strong seasonality in such INPs, with two order of magnitude higher concentrations observed during the summer. They also suggested INPs measured at this temperature may be limited most of the year by microphysical processing during transit. Stopelli et al. (2015) verified this removal mechanistic process through INP measurements and isotopic composition of fresh fallen snow at JFJ, concluding that warm temperature (i.e., INPs active at > –10 ˚C) are rapidly depleted by precipitating clouds at lower elevations. Stopelli et al. (2016) expanded their INP analyses to 2-years of data at JFJ, concluding that a high abundance of INPs at –8 ˚C is to be expected whenever high wind speed coincides with air masses having experienced little or no precipitation prior to sampling, yet a separate study by Stopelli et al. (2017) found that only a small fraction of the INPs were bacterial cells. In contrast, Lacher et al. (2018a; 2018b) conducted an interannual synopsis of INP measurements at JFJ and found anthropogenic influence on INP concentrations, but only during boundary layer influences and at relatively cold temperatures (i.e., approximately –30 ˚C), and higher INP concentrations during Saharan dust events (SDEs) and marine boundary layer air arriving at JFJ. Eriksen Hammer et al. (2018) characterized ice particle residuals and concluded that silica and aluminosilicates were the most important ice particle residuals at JFJ during the mixed-phase cloud events during Jan – Feb 2017, while carbon-rich particles of possible biological origin were of minor contribution.

Here, we demonstrate how variable sources influence INP populations depending on air mass transport and storm direction, and spectral modality between the rime, snow, and aerosols can help explain the exchange of INPs from air into cloud then into precipitation. Our results expand upon previous studies by evaluating INPs via a combination of aerosol, rime, and snow, and at a temperature range that comprises common biological and mineral INPs.

## 2 Methods

### 2.1 Aerosol, cloud rime and snow collection at Jungfraujoch

Collocated collection of snow, cloud rime, and aerosol samples for Ice Nucleation Characterization in the Alps of Switzerland (INCAS) study took place 15 Feb – 11 Mar 2018 in the Sphinx observatory at JFJ (46.55 °N, 7.98 °E; 3580 m above sea level (m a.s.l.); https://www.hfsjg.ch/en/home/). Snow was collected as described by Stopelli et al. (2015) using a Teflon-coated tin (0.1 m$^2$, 8 cm deep) for a duration of 1 – 18 hours, but typically for 1 – 4 hours. Cloud rime was collected using a slotted plexiglass plate placed vertically during snow sample collection (Lacher et al., 2017; Mignani et al., 2018). Daily size-resolved aerosol samples were collected using a Davis Rotating-drum Universal-size-cut Monitoring (DRUM) single-jet impactor (DA400, DRUMAir, LLC.) as described by Creamean et al. (2018a) from a 1-m long inlet constructed of 6.4-mm inner diameter static-dissipative polyurethane tubing (McMaster-Carr®) leading to outside of the Sphinx and connected to a funnel covered with a loose, perforated plastic bag to prevent rimed ice build-up or blowing snow from clogging the inlet. The DRUM collected aerosol particles at four size ranges (0.15 – 0.34, 0.34 – 1.20, 1.20 – 2.96, and 2.96 – >12 µm in diameter) and sampled at 27.7 L min$^{-1}$, equalling 39888 total L of air per sample. Such size ranges cover a wide array of aerosols—particularly those that serve as INPs (DeMott et al., 2010; Fridlind et al., 2012; Mason et al., 2016)—while the large volume of air collected promotes collection of rarer, warm



temperature biological INPs, but may represent a lower fraction of overall INP concentrations (Mossop and Thorndike, 1966).
Samples were deposited onto 20 x 190 mm strips of petrolatum-coated (100%, Vaseline®) perfluoroalkoxy plastic (PFA, 0.05 mm
thick) substrate secured onto the rotating drums (20 mm thick, 60 mm in diameter) in each of the four stages at the rate of 7 mm
per day (5 mm of sample streaked onto the PFA followed by 2 mm of blank).

## 2.2 Ice nucleation measurements

All samples were analysed immediately after collection for INPs using a drop freezing cold plate system described by Creamean
et al. (2018b). Briefly, snow and cloud rime samples were melted into covered 50-mL glass beakers for analysis, resulting in
approximately 10 mL of liquid per sample. Samples were manually shaken prior to analysis. Aerosols deposited onto the PFA
were prepared for drop freezing by cutting out each daily sample and placing in a 50-mL glass beaker with 2 mL of molecular
biology reagent grade water (Sigma-Aldrich®). Beakers were covered and shaken at 500 rpm for 2 hours (Bowers et al., 2009). In
between sampling, beakers were cleaned with isopropanol (99.5%), sonicated with double-distilled water for 30 minutes, then
heated at 150 °C for 30 minutes.
Copper discs (76 mm in diameter, 3.2 mm thick) were prepared by sonicating in double-distilled water for 30 minutes, cleaning
with isopropanol, then coated with a thin layer of petrolatum (Tobo, 2016; Bowers et al., 2009; Polen et al., 2018). Following
sample preparation, a sterile, single-use syringe was used to draw 0.25 mL of the suspension and 100 drops were pipetted onto the
petrolatum-coated copper disc, creating an array of ~2.5-μL aliquots. Drops were visually inspected for size; however, it is possible
not all drops were the same exact volume, which could lead to a small level of indeterminable uncertainty. However, previous
studies have demonstrated that drop size variability within this range does not significantly impact freezing results (Hader et al.,
2014b; Bigg, 1953; Langham and Mason, 1958; Creamean et al., 2018b). The copper disc was then placed on a thermoelectric cold
plate (Aldrich®) and covered with a transparent plastic dome. Small holes in the side of the dome and copper disc permitted
placement of up to four temperature probes using an Omega™ thermometer/data logger (RDXL4SD; 0.1 °C resolution and
accuracy of ± (0.4% + 1 ºC) for the K sensor types used). During the test, the cold plate was cooled at 1 – 10 °C min$^{-1}$ from room
temperature until around −35 °C. Control experiments at various cooling rates within this range show no discernible dependency
of drop freezing on cooling rate (Creamean et al., 2018b), akin to previous works (Wright and Petters, 2013; Vali and Stansbury,

138    1966).

A +0.33 °C correction factor was added to any temperature herein and an uncertainty of 0.15 °C was added to the probe accuracy
uncertainty based on DFCP characterization testing presented in Creamean et al. (2018b), to account for the temperature difference
between the measurement (i.e., in the plate centre) and actual drop temperature. Frozen drops were detected visually, but recorded
through custom software, providing the freezing temperature and cooling rate of each drop frozen. The test continued until all 100
drops were frozen. Each sample was tested three times with 100 new drops for each test. From each test, the fraction frozen and
percentage of detected frozen drops were calculated (typically, > 90% of the drops were detected). The results from the triplicate
tests were then binned every 0.5 °C to produce one spectrum per sample. Normalized differential INP spectra were created by a
using a combination of calculations. First, cumulative INP spectra were calculated using the equation posed by Vali (1971):
$$[INPs(T)](L^{-1}) = \frac{\ln N_o - \ln N_u(T)}{V_{drop}}$$



where $N_o$ is the total number of drops, $N_u(T)$ is the number of unfrozen drops at each temperature, and $V_{drop}$ is the average volume
of each drop. Aerosol INP concentrations were corrected for the total volume of air per sample ($INPs \times \frac{V_{suspension}}{V_{air}}$) while melted
rime/snow residual INPs were adjusted to the total used during analysis ($INPs \times V_{suspension}$), where $V_{suspension}$ and $V_{air}$ represent
the total liquid volume analyzed per sample (0.75 mL for the three tests) and total volume of air drawn per sample (39888 L),
respectively. Second, differential values were calculated from each 0.5-°C cumulative concentration bin. Differential INP spectra
were used early in earlier studies (Vali, 1971; Vali and Stansbury, 1966), however, spectra only reached a minimum of –20 ˚C,
missing the tail end of what are usually the highest INP concentrations as discussed in more detail below. A recent study by Polen
et al. (2018) recommend the use of differential spectra. Third, differential concentrations were divided by the maximum
concentration per sample (i.e., to normalize). Last, spectra were smoothed using a moving average.
**2.3 Supporting meteorological and source analysis data**
Auxiliary surface meteorological observations, including but not limited to hourly mean air temperature measured 2 m above
ground level (a.g.l.) (˚C), relative humidity measured 2 m a.g.l. (%), scalar wind speed (m s$^{-1}$) and direction (degrees), and incoming
longwave radiation (W m$^{-2}$) were acquired from MeteoSwiss (https://gate.meteoswiss.ch/idaweb/). From the longwave
measurements, in-cloud conditions were determined by calculating the sky temperature and comparing to air temperature measured
at the station, per the methodology of Herrmann et al. (2015) from a 6-year analysis of JFJ observations. For the current work,
each hourly measurement was categorized as out-of-cloud or in-cloud based on such calculations.
Radon ($^{222}$Rn) concentrations have been continuously measured at JFJ since 2009. Details on the detectors themselves and the
measurements can be found in Griffiths et al. (2014). Briefly, 30-minute radon concentrations were measured using a dual-flow-
loop two-filter radon detector as described by Chambers et al. (2016). Calibrated radon concentrations were converted from activity
concentration at ambient conditions to a quantity which is conserved during an air parcel's ascent: activity concentration at standard
temperature and pressure (0 ˚C, 1013 hPa), written as Bq m$^{-3}$ STP (Griffiths et al., 2014). Time periods with boundary layer
intrusion were classified as radon concentrations > 2 Bq m$^{-3}$ (Griffiths et al., 2014). Particle concentrations from approximately
0.3 to > 20 μm in diameter were measured with a 15-channel optical particle sizer (OPS 3300; TSI, Inc.) at a 1-minute time
resolution (Bukowiecki et al., 2016). Due to operational complications, OPC data were not collected prior to 23 Feb during INCAS.
Air was drawn through a heated total aerosol inlet (25 ˚C) which, besides aerosol particles, enables hydrometeors with diameters
< 40 mm to enter and to evaporate, at wind speeds of 20 m s$^{-1}$ (Weingartner et al., 1999). SDEs were determined from existing
methodology using various aerosol optical properties, but specifically, the Ångström exponent of the single scattering albedo
(å$_{SSA}$), which decreases with wavelength during SDEs (Collaud Coen et al., 2004; Bukowiecki et al., 2016). SDEs are automatically
detected by the occurrence of negative å$_{SSA}$ that last more than four hours. Most of the SDEs do not lead to a detectable increase
of the 48-h total suspended particulate matter (TSP) concentrations at JFJ. Additionally, we consider these events probable SDEs,
but may have influences from other sources in addition.
Air mass transport analyses were conducted using the HYbrid Single Particle Lagrangian Integrated Trajectory model with the
SplitR package for RStudio (https://github.com/rich-iannone/SplitR) (Draxler, 1999; Draxler and Rolph, 2011; Stein et al., 2015).
Reanalysis data from the National Centers for Environmental Prediction (NCEP) National Center for Atmospheric Research
(NCAR) (2.5˚ latitude-longitude; 6-hourly; https://www.ready.noaa.gov/gbl_reanalysis.php) were used as the meteorological fields
in HYSPLIT simulations. Trajectories were initiated at 10, 500, and 1000 m a.g.l. every 3 hours daily. Trajectories were only



## 3 Results and discussion

### 3.1 Directional dichotomy of storm systems during INCAS

Local surface meteorology was variable at JFJ during INCAS, with air temperatures ranging from –27.5 to –4.8 ˚C (average of –13.7 ˚C)—temperatures relevant to heterogeneous nucleation of cloud ice—and relative humidity ranging from 18 to 100% (Figure 1a). Wind speed was 6.4 m s$^{-1}$ on average, with spikes during most storm systems up to 22.8 m s$^{-1}$ (i.e., wind speed during rime and snow collection; Figure 1b). Due to the topography surrounding JFJ, predominant wind directions were northwest followed by southeast, with the fastest winds recorded originating from the southeast (Figure 2). Such conditions are typical for JFJ during the winter (Stopelli et al., 2015). Out of the entire study, several days were classified as northwesterly or southeasterly during storm conditions when a combination of aerosol, cloud rime, and snow samples were collected (i.e., a full 24 hours of northwesterly or a full 24 hours of southeasterly winds during snowfall; Table 1), which are herein focused on as the case study days.

Extending past local conditions, air mass transport 10 days back in time prior to reaching JFJ on case study days was, as expected, dissimilar between northwesterly (Figure 3) and southeasterly (Figure 4) conditions. The main distinctions between northwesterly and southeasterly days are: (1) northwesterly days originated from farther west, with some days reaching back to the Canadian Archipelago, while air masses on southeasterly days predominantly hovered over land and occasional oceanic sources closer to Europe, (2) southeasterly air masses travelled closer to the surface relative to northwesterly days, especially south and east of JFJ while northwesterly air masses were typically transported from higher altitudes (i.e., more free tropospheric exposure), and (3) aside from 06 Mar (which is discussed in more detail in the following section), northwesterly air masses did not travel over the Mediterranean and northern Africa, whereas the southeasterly air masses reaching down to 100 m above JFJ arrived from over such regions within less than 2 days before arriving to JFJ. Boose et al. (2016) reported similar transport pathways for JFJ during multiple consecutive winters and concluded that marine and Saharan dust served as dominant sources of INPs. Reche et al. (2018) also reported similar pathways and sources for bacteria and viruses, but during the summer in southern Spain. Possible SDEs were automatically detected on 24 Feb and 10 Mar in the current work, and air mass transport pathways are shown for these days. These disparate sources and transport pathways of air support the variability in the ice nucleation observations as discussed in more detail in the following section.

As evidenced by the air mass transport analyses, each southeasterly case day (and 06 Mar) experienced longer residence times in what was likely the boundary layer (i.e., 1000 m or less) compared to northwesterly cases, which is supported by $^{222}$Rn data (Figure 5). Griffiths et al. (2014) determined that radon concentrations > 2 Bq m$^{-3}$ signify boundary layer intrusion, which in the current work was clearly observed on 23 Feb, 06 Mar, and 11 Mar, indicating samples collected on these days were likely influenced by boundary layer sources (planetary and marine). Relatively low radon concentrations were observed the remaining case study days, indicating these samples were predominantly affected by free tropospheric air and thus, lower aerosol concentrations and/or more distant sources. Although OPC data were missing until 23 Feb, source information can be gleaned from the available data. For example, 23 Feb had episodic high concentrations of particles (maximum of 9.6 cm$^{-3}$) towards the beginning of the day coincident with the largest spike in radon, with a steady decrease as time transpired, indicating the boundary layer was an ample source of > 0.3 μm particles. Although not a case study time period, a similar correlation between the OPC and radon concentrations was observed 27 – 28 Feb, where the highest concentrations of each were observed during the entire study time period. Selected days



were subject to diurnal upslope winds (Figure 1b), such as 6 Mar, where boundary layer air reached JFJ and a midday maximum
in OPC particle concentrations was observed, indicating lower elevations were the dominant source of aerosol. Although, diurnal
variations in aerosol from local sources have been shown to not be common in the winter at JFJ (Baltensperger et al., 1997). In
contrast, 11 Mar was exposed to boundary layer air based on radon observations, but particle concentrations were low (average of
0.2 cm$^{-3}$ compared to a study average of 3.0 cm$^{-3}$), signifying that although boundary layer intrusion occurred at JFJ, it was not a
substantial source of aerosol. These relationships corroborate the ice nucleation observations, as discussed in detail below.

**3.2 Variability in INP spectral properties based on storm characteristics**

Out of the 25 aerosol, 30 rime, and 39 snow samples collected, 7 aerosol, 19 rime, and 23 snow were collected northwesterly or
southeasterly case study days (Table 1). Mixed wind direction days were excluded, as sources from both directions would
contribute to the daily aerosol sample. Figure 6 shows INP concentrations from aerosol, snow, and rime samples on the case days
compared to air temperature, wind speed, and previous measurements at JFJ. Only the largest size range of the aerosol is shown
because the remaining size ranges (i.e., < 2.96 μm) were not distinct with respect to wind direction. The fact that size, alone,
exhibited directionally-dependent results and that such dependencies were only observed in the coarse mode aerosol indicate: (1)
the sources were indeed different between northwesterly and southeasterly transport—supporting the air mass source analyses—
and (2) the coarse mode aerosols were likely from a regional source as opposed to long-range transported thousands of kilometres.
This is because gravitational settling typically renders transport of coarse particles inefficient especially within the boundary layer
(Creamean et al., 2018a). Previous work by Collaud Coen et al. (2018) concludes that the local boundary layer never influences
JFJ in the winter, supporting the fact that regional sources were likely prominent in the current work.
Generally, INPs from southeasterly days were higher in concentration and more efficient (i.e., were warm temperature INPs that
facilitated ice formation > –15 ℃) than northwesterly samples. Our results are parallel to those by Stopelli et al. (2016), who also
observed higher INP concentrations in snow samples collected during southerly conditions at JFJ from Dec 2012 to Oct 2014.
However, when comparing overlapping temperature ranges from the snow samples during the winter only (Figure 6a),
concentrations reported here are generally higher than those reported by Stopelli et al. (2016), especially at the highest
temperatures. Aside from some of the snow samples, onset freezing temperatures (i.e., the highest temperature in which each
sample froze) were higher for samples from the southeast as compared to the northwest (Figure 6b), indicating more efficient INPs
from the southeast. However, a larger (smaller) spread in onset temperatures was observed in samples from the northwest
(southeast), suggesting two possibilities: (1) influences were more (less) variable sources from the northwest (southeast), as
discussed in more detail in the following section and/or (2) in the case of cloud rime and snow, clouds from the northwest were
already depleted with the most efficient INPs due to precipitation prior to arriving at JFJ (i.e., higher transport altitudes which
could have been exposed to cloudy conditions as compared to the southeast days which exhibited transport closer to the ground;
Figures 3 and 4).
There was no clear correlation between INP concentrations with air temperature but air temperatures tended to be higher for
northwesterly as compared to southeasterly cases. At –25 ℃ freezing temperatures for the INPs, most northwesterly samples had
a range of INP concentrations at higher air temperatures (i.e., > –9 ℃), while southeasterly samples exhibited overall higher INP
concentrations, but still at a range of air temperatures (Figure 6e). In contrast, there was no correlation or gradient relationship
between INP concentrations at any temperature and wind speed (Figure 6f – h), unlike the correlation between wind speed and
INPs at –8 ℃ observed by Stopelli et al. (2016). We also evaluated INP concentrations versus wind speed at –8 ℃ but did not see
any correlation (not shown). Regarding the snow, it is possible that surface processes generate airborne ice particles, which





contribute to a snow sample collected at a mountain station (Beck et al., 2018). However, snow that is re-suspended during a
snowfall event largely consists of the most recently fallen snow crystals covering wind-exposed surfaces. These particles are
unlikely to be different from concurrently falling snow. Hence, their contribution will not change INP abundance or spectral
properties of the collected sample. Another matter are hoar frost crystals, which can be very abundant in terms of number, but
because of their small size (i.e., < 100 µm (Lloyd et al., 2015)) can only make a minor contribution to the mass of solid precipitation
depositing in a tin placed horizontally on a mountain crest. The majority of small crystals will follow the streamlines of air passing
over the crest. All that an increased influence of hoar frost particles would do to our observations is to decrease measured
differences between snow and rime samples.
Figure 7 shows the cumulative and normalized differential INP spectra from the northwesterly and southeasterly case day samples.
In addition to containing higher concentrations of warm temperature INPs, more southeasterly samples contained a bimodal
distribution relative to the colder and unimodal distributions from northwesterly samples. The warm mode, or "bump" at
temperatures above approximately –20 °C has been observed in a wide range of previous immersion mode ice nucleation studies
including but not limited to some of the earliest studies of total aerosol (Vali, 1971), residuals found in hail (Vali and Stansbury,
1966), sea spray aerosol (McCluskey et al., 2017; DeMott et al., 2016), soil samples (Hill et al., 2016), agricultural harvesting
emissions (Suski et al., 2018), controlled laboratory measurements of mixtures of bacteria and illite (Beydoun et al., 2017), and in
recent reviews of aerosol (Kanji et al., 2017; DeMott et al., 2018) and precipitation (Petters and Wright, 2015) samples. Most
previous studies that show spectra with the warm mode typically: (1) report a wide range of freezing temperatures such that it can
be observed relative to the steady increase of INPs at colder temperatures (i.e., Figure 7, left column) or (2) are of samples that
include a mixture of biological and mineral or other less efficient INP sources. For example, several previous studies report INP
concentrations down to only –15 °C (e.g., Conen and Yakutin, 2018; Hara et al., 2016; Kieft, 1988; Schnell and Vali, 1976; Vali
et al., 1976; Wex et al., 2015), namely because the goal was to target efficient, warm-temperature biological INPs. However, the
warm mode may not be evident in such studies, given it cannot be visualised next to the drastic increase in INPs with temperatures
below –15 °C (i.e., the cold mode). In contrast, studies conducting INP measurements on known mineral dust samples also are not
able to observe the warm mode (e.g., Price et al., 2018; Atkinson et al., 2013; Murray et al., 2012). Recent laboratory and modelling
work by Beydoun et al. (2017) demonstrates the transition between warm and cold mode INPs using controlled experiments of
Snowmax and illite mixtures. Together, it is apparent that a mixed biological and mineral (or less efficient biological INPs) sample
is needed to assess the bimodal behaviour in the INP spectra.
**3.3 Potential components of INP populations at JFJ**
Taking the spectral characteristics in the context of air mass transport can help elucidate the possible sources of INPs at JFJ during
INCAS. Qualitative and quantitative evaluation of the warm mode, or likely, the relative contribution of warm temperature
biological INPs to cold mode INPs is transparent when differential INP spectra are calculated (Figure 7). Additionally, normalizing
such spectra affords a qualitative comparison of spectra signatures between aerosols and residuals found in cloud rime and snow.
Figure 8 shows normalized differential spectral characteristics of daily aerosol, rime, and snow INPs. We offer some possible
explanations for the observed variability between the samples. Naturally, the boundary layer more frequently than not contains
higher concentrations of warm temperature INPs—and INPs in general—as compared to the free troposphere given the proximity
to the sources (e.g., forests, agriculture, vegetation, and the oceans) (Burrows et al., 2013; Despres et al., 2012; Frohlich-Nowoisky
et al., 2016; Wilson et al., 2015; Burrows et al., 2009a; Burrows et al., 2009b; Frohlich-Nowoisky et al., 2012; Suski et al., 2018).



Although, microorganisms and nanoscale biological fragments are episodically lofted into the free troposphere with mineral dust
and transported thousands of kilometres (Creamean et al., 2013; Kellogg and Griffin, 2006).
Air arriving at JFJ on 15 and 16 Feb originated from the farthest away and were not heavily exposed to boundary layer air, as
evidenced by the air mass trajectory analysis (Figure 3) and radon (Figure 5), indicating long-range transport in the free
troposphere. This could explain why the warm mode was observed in the rime and snow, but not the aerosol—the aerosol had
sufficient time to nucleate ice during free tropospheric transport and especially the warm temperature INPs that would likely
become depleted in-cloud first (Stopelli et al., 2015), as also supported by the higher INP concentrations in most of the rime and
snow compared to the aerosol in Figure 6b. Cloud fraction was relatively low (12.5 to 25%), but air temperatures were relatively
high (–8.4 to –7.1 ℃), suggesting conditions were amenable for long-range transported warm temperature INPs to nucleate cloud
ice. However, from the available data, we cannot determine with certainty if the local conditions were the same as those when
nucleation initially occurred. For 19 and 20 Feb, air temperature was very cold (–16.4 and –19.6 ℃, respectively) cloud fraction
was high (92 and 54%, respectively), and all samples remained unimodal (i.e., only containing the cold mode). One possible
explanation is that any warm temperature INPs that were present in the clouds had already snowed out prior to reaching the
sampling location, as observed by Stopelli et al. (2015) at JFJ. Although, given the low radon concentrations and erratic transport
pathways, it is possible such air masses did not contain a relatively large concentrations of warm temperature INPs due to deficient
exchange with the boundary layer. It was not until the southeasterly cases that the aerosol samples exhibited bimodal characteristics
(i.e., contained both the warm and cold modes). Specifically, on 23 Feb local winds shifted to southeasterly (147 degrees on
average) and air masses arrived from over the eastern Alps, Eastern Europe, Scandinavia, and earlier on in time, the Atlantic Ocean.
Thus, these samples were predominantly influenced by the continental (mostly over remote regions) and marine boundary layers
(Figures 4 and 5), where sources of warm temperature INPs are more abundant (Frohlich-Nowoisky et al., 2016). The northwesterly
case of 06 Mar is somewhat interesting in that the local wind direction was clearly from the northwest, but air mass source analyses
show transport in the boundary layer (radon) from the south, when looking farther back in time, traveling over the Mediterranean
and North Africa. The aerosol sample had the third highest onset temperature for INPs (Figure 6b) and snow samples exhibited
bimodality (Figure 8c). It is the only one of the northwesterly case samples that encountered boundary layer exposure according
to the radon observations. Combined, these results suggest a somewhat mixed-source sample, and that 06 Mar may not be directly
parallel to the other northwesterly cases. Transitioning back to a southeasterly case on 11 Mar, only the rime and snow unveiled
bimodal behaviour from air transported from similar regions as the 06 Mar sample. However, transport on 11 Mar was more
directly from the south over the Mediterranean and North Africa, indicating less time for removal of the INPs during transport.
Additionally, OPC concentrations were very low (Figure 5). These results suggest the aerosols already nucleated cloud ice prior
to reaching JFJ, which is supported by the 10 Mar sampling where the aerosol was bimodal, rime was unimodal, and snow was
bimodal, but the warm mode resided at a relatively cold temperature (–16.5 ℃).
Two other days without snowfall support the conclusion that southeasterly air mass transport introduces warm temperature INPs
to JFJ. On 24 Feb, clouds were present at JFJ (a cloud fraction of 37.5%), but riming was insufficient to collect a sufficient quantity
for INP analysis and no snowfall occurred. Interestingly, the warm mode was the maximum for the aerosol sample—normally, the
cold mode has the highest normalized value—indicating a larger contribution of warm temperature INPs as compared to the total
INP population. Air mass transport was very similar to 23 Feb signifying similar INP sources, but it is probable that a slightly
warmer (–6.0 as compared to –9.8 ℃ air temperature), drier (79 versus 89% relative humidity), and higher pressure (649 versus
645 mb) postfrontal system moved over JFJ on 24 Feb, inhibiting removal of warm temperature INPs during transport relative to
the day prior. The second case, 28 Feb (not shown) was very similar to 24 Feb in that: (1) only an aerosol sample was collected





and (2) the warm mode was the maximum mode. As compared to 27 Feb where a warm mode was not observed, 28 Feb was
warmer (−20.0 as compared to −26.2 ˚C), drier (52 versus 62%), higher pressure (635 versus 630 mb), and had a warmer onset
temperature (–6.8 versus –14.8 ˚C). Wind direction was slightly different: southeasterly (153 degrees) on 27 Feb as compared to
southwesterly on 28 Feb (221 degrees). However, conditions were cloudier than the 23 – 24 Feb coupling and completely cloudy
on 27 Feb (100 and 66.7% cloud fraction on 27 Feb and 28 Feb, respectively). Additionally, radon and OPC concentrations were
the highest on 27 – 28 Feb as compared to the rest of the days during INCAS (Figure 5). Combined, these results suggest a very
local, boundary layer source of INPs started on 27 Feb, but were quickly depleted in the very cloudy conditions. Once clouds
started to clear and a shift in frontal characteristics occurred, a similar source of very efficient warm temperature INPs affected JFJ
but were able to be observed in the aerosol.
**4 Conclusions**
Aerosol, cloud rime, and snow samples were collected at the High Altitude Research Station Jungfraujoch during the INCAS field
campaign in Feb and Mar 2018. The objectives of the study were to assess variability in wintertime INP sources found in cloudy
environments and evaluate relationships between INPs found in the different sample materials. To directly compare air to liquid
samples, characteristics of normalized differential INP spectra were compared in the context of cumulative INP spectra statistics,
air mass transport and exposure to boundary layer or free tropospheric conditions, and local meteorology. Distinction between
northwesterly and southeasterly conditions yielded variable results regarding INP efficiency and concentrations, biological versus
non-biological sources, and meteorological conditions at the sampling location. In general, INP concentrations were 3 to 20 times
higher for all sample types when sources from the southeast infiltrated JFJ, while the modality of the INP spectra was bimodal for
aerosol but variable for the rime and snow depending on meteorological context.
In general, comprehensive measurements of INPs from aerosol, and rime and snow when possible, affords useful information to
compare and explain exchange between aerosols, clouds, and precipitation in the context of local and regional scale meteorology
and transport conditions. Assessment of INP modality and spectra statistics adds another dimension for qualitative and semi-
quantitative intercomparison of sampling days and evaluation of associations between aerosol, cloud, and precipitation sampling.
Extending INP analyses past reporting cumulative concentrations affords more detailed information on the population of INPs and
enables comparison between samples from aerosols, clouds, precipitation, and beyond (e.g., seawater, soil, etc.). Using auxiliary
measurements and air mass simulations, in addition to context provided by previous work at JFJ, we have addressed possible
sources for INCAS. However, more detailed source apportionment work should be imminent to comprehensively characterize INP
sources based on spectral features. Future studies should ideally use such statistical analyses in tandem with focused chemical and
biological characterization assessments to provide direct linkages between INP spectral properties and sources. Such investigations
could yield valuable information on INP sources, and aerosol-cloud-precipitation interactions, which could then be used to improve
process-level model parameterizations of such interactions by rendering quantitative information on INP source, efficiency, and
abundance. Improving understanding of aerosol impacts on clouds and precipitation will ultimately significantly enhance
understanding of the earth system with respect to cloud effects on the surface energy and water budgets to address future concerns
of climate change and water availability.
**Author contributions**



JMC collected the samples, conducted the DFA sample analysis, conducted data analysis, and wrote the manuscript. CM and FC
also contributed to collecting rime and snow samples. JMC, CM, and FC designed the experiments. NB provided quality controlled
OPS data. CM, NB, and FC helped with manuscript feedback and revision prior to submission.

**Acknowledgements**

The Swiss National Science Foundation (SNF) financially supported JMC within its Scientific Exchanges Programme, through
grant number IZSEZ0_179151. The work of CM and FC on Jungfraujoch is made possible through SNF grant number
200021_169620. Aerosol data were acquired by Paul Scherrer Institute in the framework of the Global Atmosphere Watch (GAW)
programme funded by MeteoSwiss. Further support was received from the ACTRIS2 project funded through the EU H2020-
INFRAIA-2014-2015 programme (grant agreement no. 654109) and the Swiss State Secretariat for Education, Research and
Innovation (SERI; contract number 15.0159-1). The opinions expressed and arguments employed herein do not necessarily reflect
the official views of the Swiss Government. We are grateful to the International Foundation High Altitude Research Stations
Jungfraujoch and Gornergrat (HFSJG), 3012 Bern, Switzerland, for providing through its infrastructure comfortable access to
mixed-phase clouds. Special thanks for go to Joan and Martin Fischer, and Christine and Ruedi Käser, the custodians of the station.
The authors gratefully acknowledge the NOAA Air Resources Laboratory (ARL) for the provision of the HYSPLIT transport and
dispersion model and/or READY website (http://www.ready.noaa.gov) used in this publication.

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




**Table 1. Dates and times for cloud rime, snow, and aerosol samples collected during the 2018 winter INCAS study at Jungfraujoch. Samples highlighted in blue and red are the northwest and southeast case studies, respectively.**

| Date | Cloud rime | | | Snow | | | Aerosol | | | |
|---|---|---|---|---|---|---|---|---|---|---|
| | Sample | Start (UTC) | Duration (hh:mm) | Sample | Start (UTC) | Duration (hh:mm) | Sample | Start (UTC) | Duration (hh:mm) | Stages |
| 15-Feb | Rime1 | 06:30 | 03:07 | Snow1 | 07:15 | 01:15 | DRUM1 | 10:00 | 12:00 | A/B/C/D |
| | Rime2 | 09:37 | 02:13 | Snow2 | 08:40 | 01:30 | DRUM2 | 22:00 | 24:00 | A/B/C/D |
| | Rime3 | 11:50 | 03:55 | Snow3 | 10:10 | 01:35 | | | | |
| | Rime4 | 15:45 | 03:35 | Snow4 | 12:45 | 02:30 | | | | |
| | Rime5 | 19:20 | 13:55 | Snow5 | 15:20 | 04:00 | | | | |
| 16-Feb | Rime6 | 07:15 | 02:10 | Snow6 | 07:15 | 02:02 | DRUM3 | 22:00 | 24:00 | A/B/C/D |
| | Rime7 | 09:29 | 02:41 | Snow7 | 09:23 | 02:37 | | | | |
| | | | | Snow8 | 14:00 | 17:50 | | | | |
| 17-Feb | Rime8 | 12:08 | 01:16 | Snow9 | 07:50 | 02:23 | DRUM4 | 22:00 | 24:00 | A/B/C/D |
| | Rime9 | 13:24 | 02:23 | Snow10 | 10:16 | 01:10 | | | | |
| | Rime10 | 15:47 | 03:12 | Snow11 | 11:35 | 00:33 | | | | |
| | Rime11 | 18:59 | 06:48 | Snow12 | 12:20 | 01:04 | | | | |
| | | | | Snow13 | 13:42 | 01:00 | | | | |
| | | | | Snow14 | 14:45 | 00:55 | | | | |
| | | | | Snow15 | 15:54 | 02:54 | | | | |
| | | | | Snow16 | 18:52 | 05:50 | | | | |
| 18-Feb | Rime12 | 00:47 | 08:22 | | *None* | | DRUM5 | 22:00 | 24:00 | A/B/C |
| 19-Feb | Rime13 | 21:00 | 10:50 | Snow17 | 21:00 | 08:50 | DRUM6 | 22:00 | 24:00 | A/B/C |
| 20-Feb | Rime14 | 05:50 | 04:14 | Snow18 | 05:50 | 04:14 | DRUM7 | 22:00 | 24:00 | A/B/C |
| | Rime15 | 12:08 | 02:17 | Snow19 | 12:08 | 02:14 | | | | |
| 21-Feb | | *None* | | | *None* | | DRUM8 | 22:00 | 24:00 | A/B/C/D |
| 22-Feb | | *None* | | | *None* | | DRUM9 | 22:00 | 24:00 | A/B/C/D |
| 23-Feb | Rime16 | 20:00 | 14:30 | Snow20 | 07:49 | 02:51 | DRUM10 | 22:00 | 24:00 | A/B/C/D |
| | | | | Snow21 | 10:55 | 03:35 | | | | |
| | | | | Snow22 | 14:40 | 02:42 | | | | |
| | | | | Snow23 | 20:00 | 12:11 | | | | |
| 24-Feb | | *None* | | | *None* | | DRUM11 | 22:00 | 24:00 | A/B/C |
| 25-Feb | | *None* | | | *None* | | DRUM12 | 22:00 | 24:00 | A/B |
| 26-Feb | | *None* | | | *None* | | DRUM13 | 22:00 | 24:00 | A/B/C |
| 27-Feb | | *None* | | | *None* | | DRUM14 | 22:00 | 24:00 | A/B/C |
| 28-Feb | | *None* | | | *None* | | DRUM15 | 22:00 | 24:00 | A/B/C |
| 01-Mar | | *None* | | | *None* | | DRUM16 | 22:00 | 24:00 | A/B/C |
| 02-Mar | | *None* | | | *None* | | DRUM17 | 22:00 | 24:00 | A/B/C |
| 03-Mar | | *None* | | | *None* | | DRUM18 | 22:00 | 24:00 | A/B/C |
| 04-Mar | | *None* | | | *None* | | DRUM19 | 22:00 | 24:00 | A/B/C |
| 05-Mar | Rime17 | 16:43 | 15:32 | Snow24 | 21:52 | 08:16 | DRUM20 | 22:00 | 24:00 | A/B/C |
| 06-Mar | Rime18 | 06:15 | 03:03 | Snow25 | 06:15 | 02:50 | DRUM21 | 22:00 | 24:00 | A/B/C |
| | Rime19 | 09:18 | 05:42 | Snow26 | 09:14 | 05:26 | | | | |
| | Rime20 | 15:00 | 02:08 | Snow27 | 14:54 | 01:56 | | | | |
| | Rime21 | 17:08 | 04:41 | Snow28 | 17:26 | 04:04 | | | | |
| | Rime22 | 22:38 | 21:40 | Snow29 | 22:38 | 07:35 | | | | |
| 07-Mar | Rime23 | 06:19 | 03:58 | Snow30 | 06:19 | 01:19 | DRUM22 | 22:00 | 24:00 | A/B/C |
| | Rime24 | 10:17 | 05:40 | Snow31 | 07:50 | 02:00 | | | | |
| | Rime25 | 15:57 | 08:31 | Snow32 | 12:49 | 02:59 | | | | |
| | Rime26 | 22:28 | 06:34 | Snow33 | 16:00 | 05:19 | | | | |
| | | | | Snow34 | 22:28 | 06:27 | | | | |
| 08-Mar | | *None* | | | *None* | | DRUM23 | 22:00 | 24:00 | A/B/C |
| 09-Mar | | *None* | | | *None* | | DRUM24 | 22:00 | 24:00 | A/B/C |
| 10-Mar | Rime27 | 11:00 | 21:46 | Snow35 | 06:00 | 04:00 | DRUM25 | 22:00 | 24:00 | A/B/C |
| | | | | Snow36 | 10:10 | 01:56 | | | | |
| 11-Mar | Rime28 | 06:46 | 02:59 | Snow37 | 05:48 | 04:03 | | *None* | | |
| | Rime29 | 09:56 | 03:49 | Snow38 | 09:54 | 03:51 | | | | |
| | Rime30 | 13:45 | 04:48 | Snow39 | 13:48 | 02:28 | | | | |





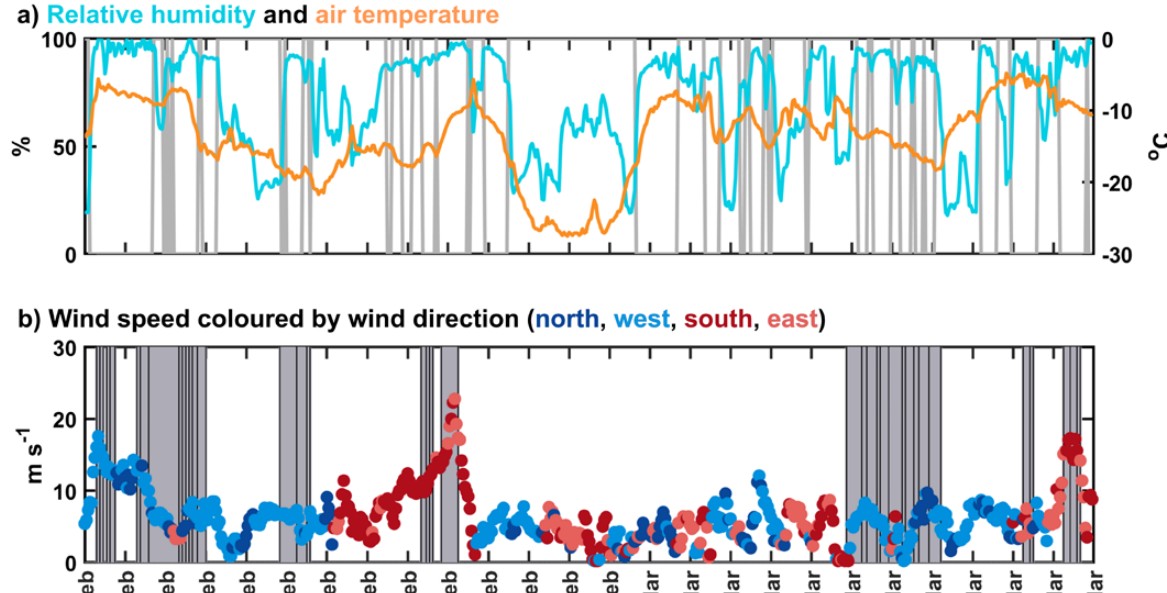


**Figure 1. Meteorological data at JFJ from INCAS, including a) relative humidity and surface air temperature and b) wind speed colored by wind direction. Grey shading in a) indicates in-cloud measurement conditions per the estimation by Herrmann et al. (2015) and in b) indicates snow and cloud rime collections with the width of the bar indicating the duration of each sample. Aerosol sampling was conducted daily during all conditions (i.e., precipitation, cloudy, and clear sky). North, east, south, and west correspond to wind direction ranges of 315 – 45, 45 – 135, 135 – 225, and 225 – 315 degrees, respectively.**




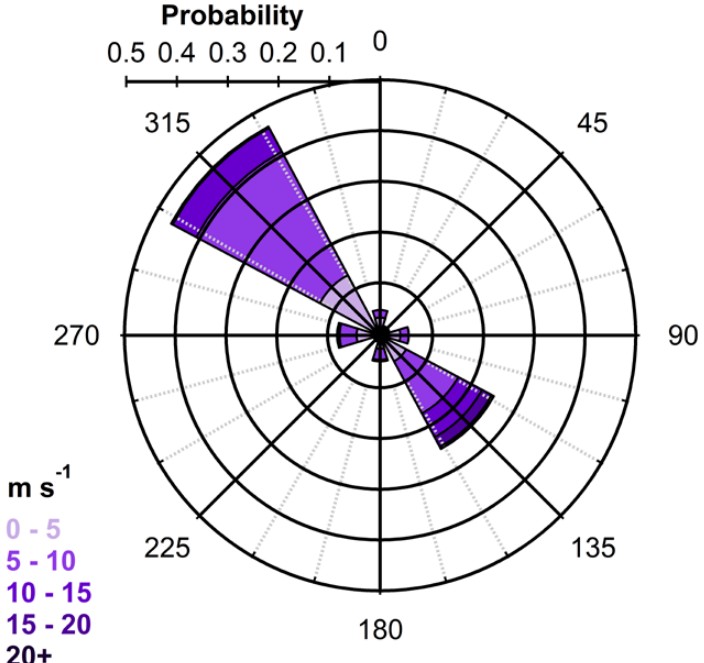

667

**Figure 2. Rose plot for wind data during INCAS. Values correspond to wind direction binned by 45 degrees and wind speeds binned by**
668

**5 m s⁻¹. The probability for wind speed to fall within these bins is plotted.**
669





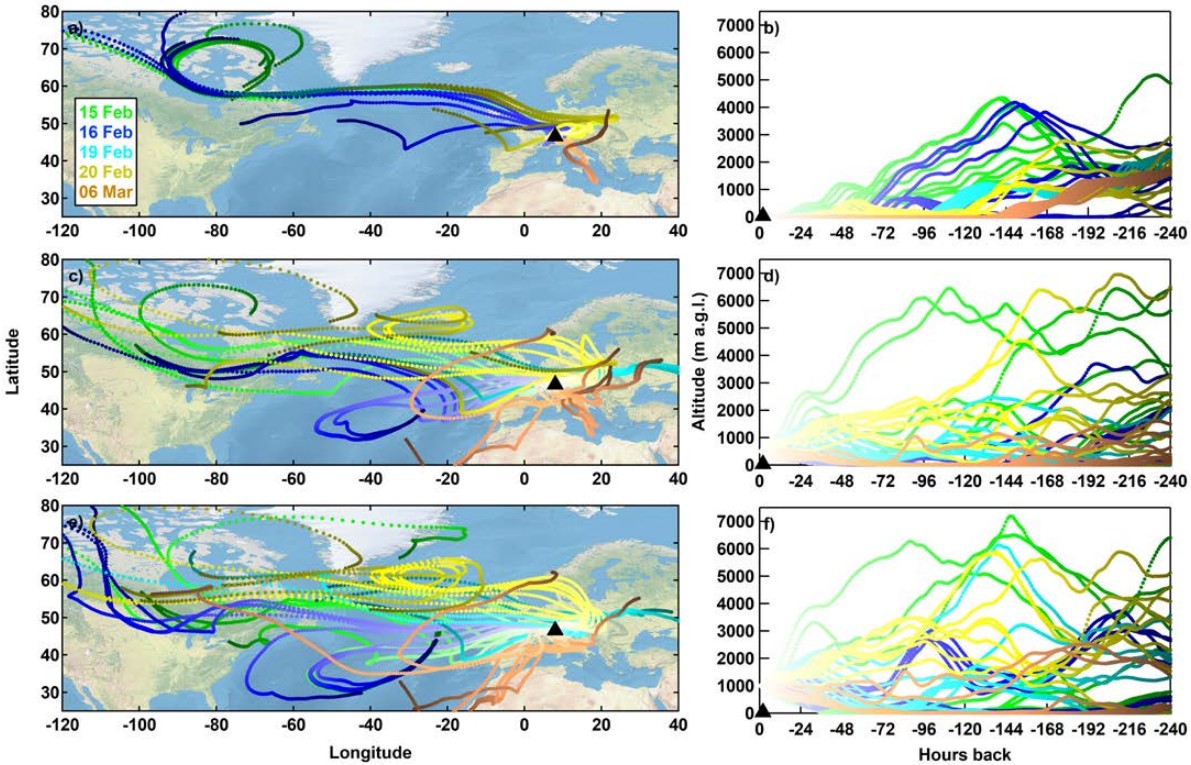

**Figure 3. 10-day air mass backward trajectories initiated every 3 hours during sample collection for northwesterly case days at a) 10, c) 500, and e) 1000 m a.g.l. Altitude profiles versus time are also shown for b) 10, d) 500, and f) 1000 m a.g.l. Each day is coloured differently to differentiate between the days.**





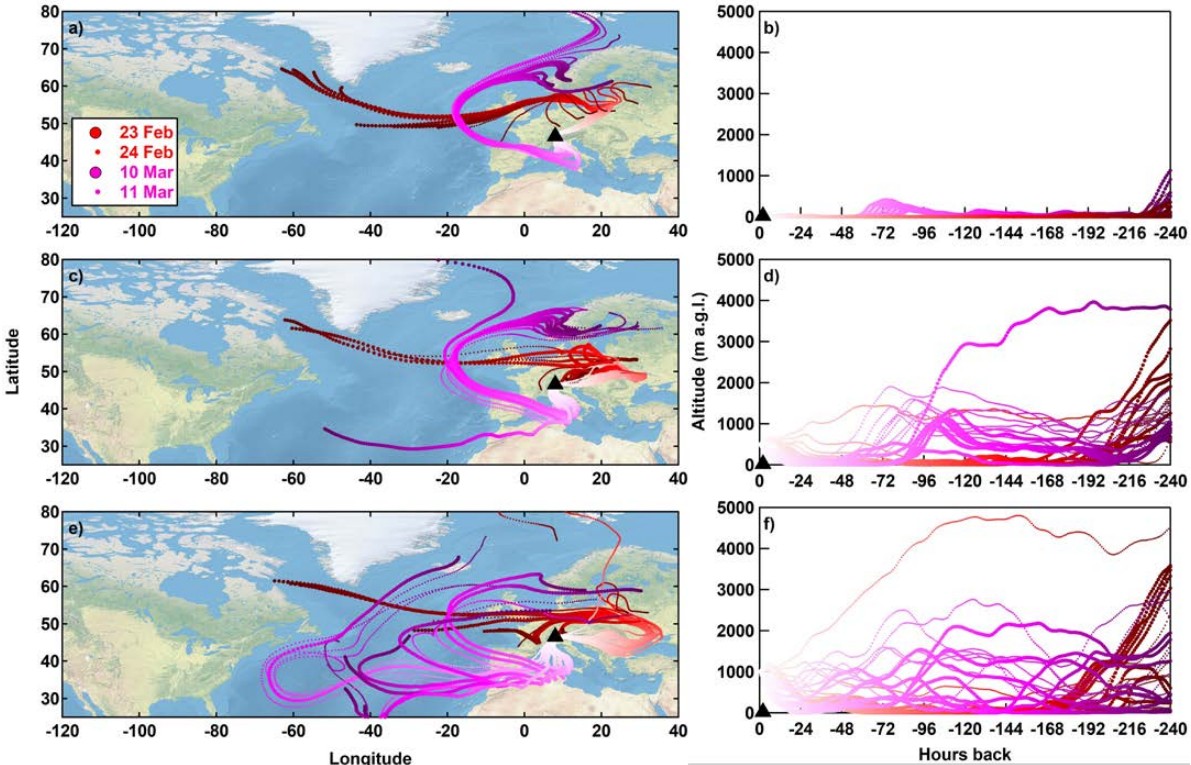


**Figure 4. Same as Figure 3, but for southeasterly case days. 24 Feb and 10 Mar are shown as smaller markers, indicative of possible Saharan dust events from Paul Scherrer Institute.**





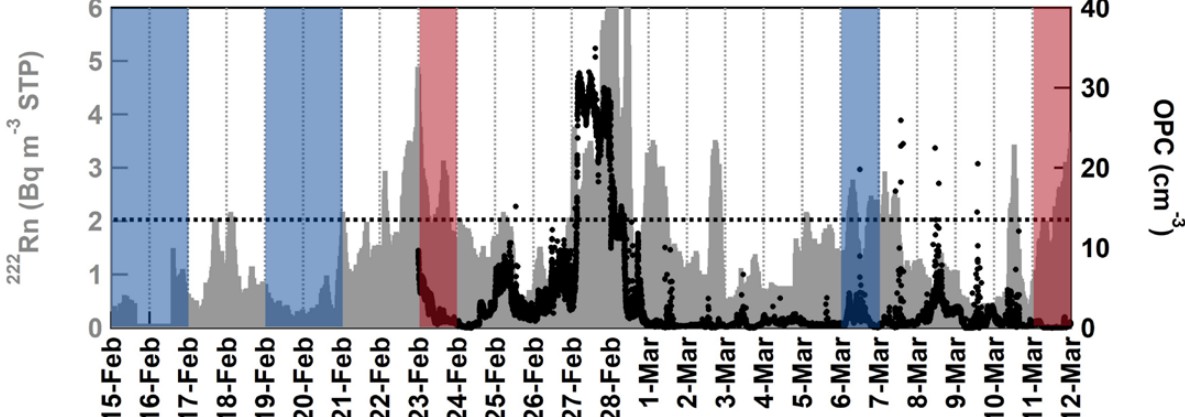

**Figure 5.** **²²²Rn concentrations (grey) measured and corrected for standard temperature and pressure during INCAS. OPC particle**
**number concentrations (black) are also shown, but data were missing prior to 23 Feb. The black dashed line indicates a threshold of 2**
**Bq m⁻³ whereby boundary layer intrusion likely occurred at JFJ. Blue and red shadings represent northwesterly and southeasterly case**
**study days, respectively.**



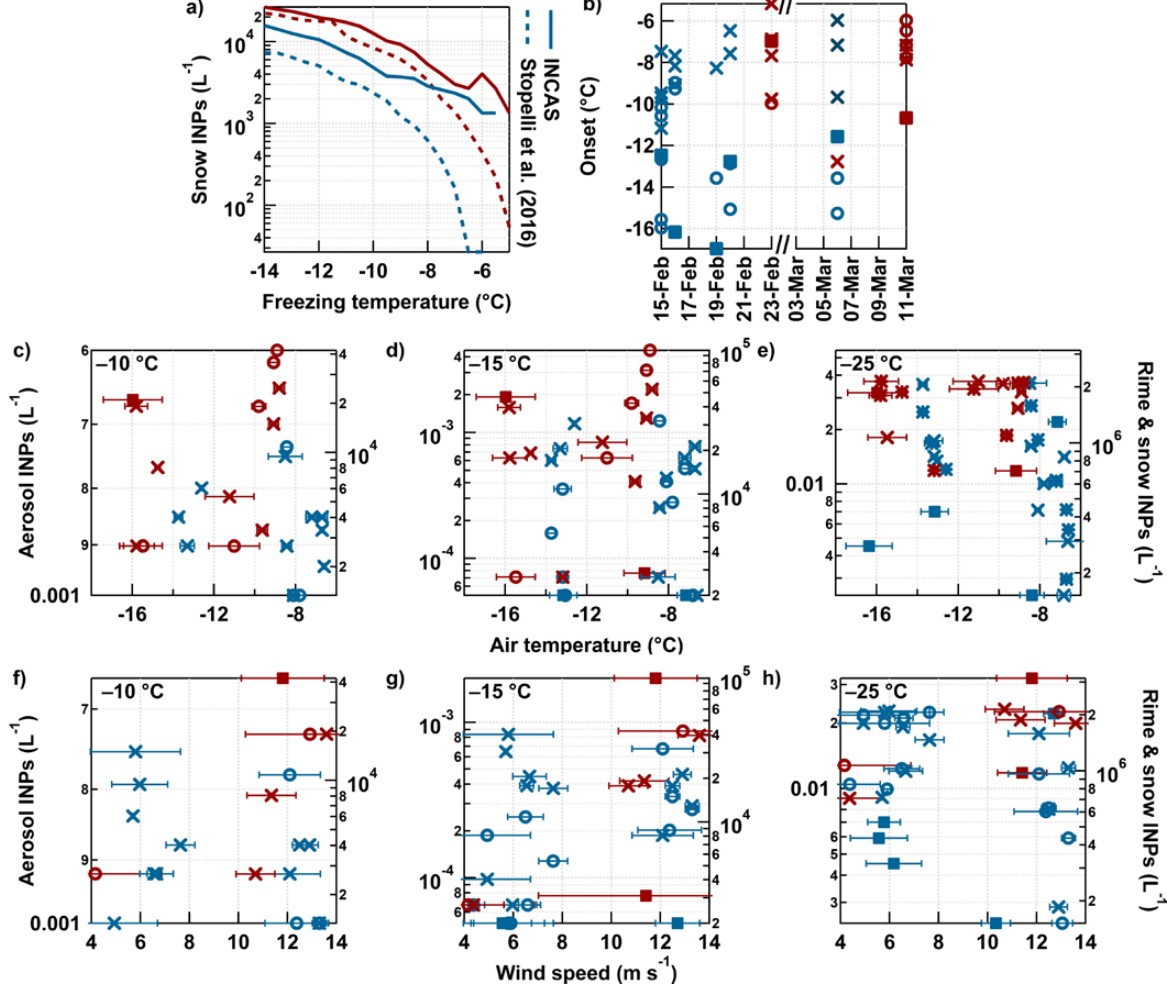


**Figure 6. a) Comparison of INCAS snow INPs within the same range of those reported by Stopelli et al. (2016) for measurements at JFJ during the 2012/13 winter season. Summary of INCAS INP concentrations from aerosol (squares), cloud rime (open circles), and snow samples (x's), including b) freezing onset temperatures, and correlations between air temperature averages during sample collection and INPs at freezing temperatures of c) −10 °C, d) −15 °C, and e) −25 °C. The same concentrations at f) −10 °C, g) −15 °C, and h) −25 °C are plotted against average wind speed measured during sample collection periods. Blue and red markers represent northwesterly and southeasterly wind directions, respectively.**





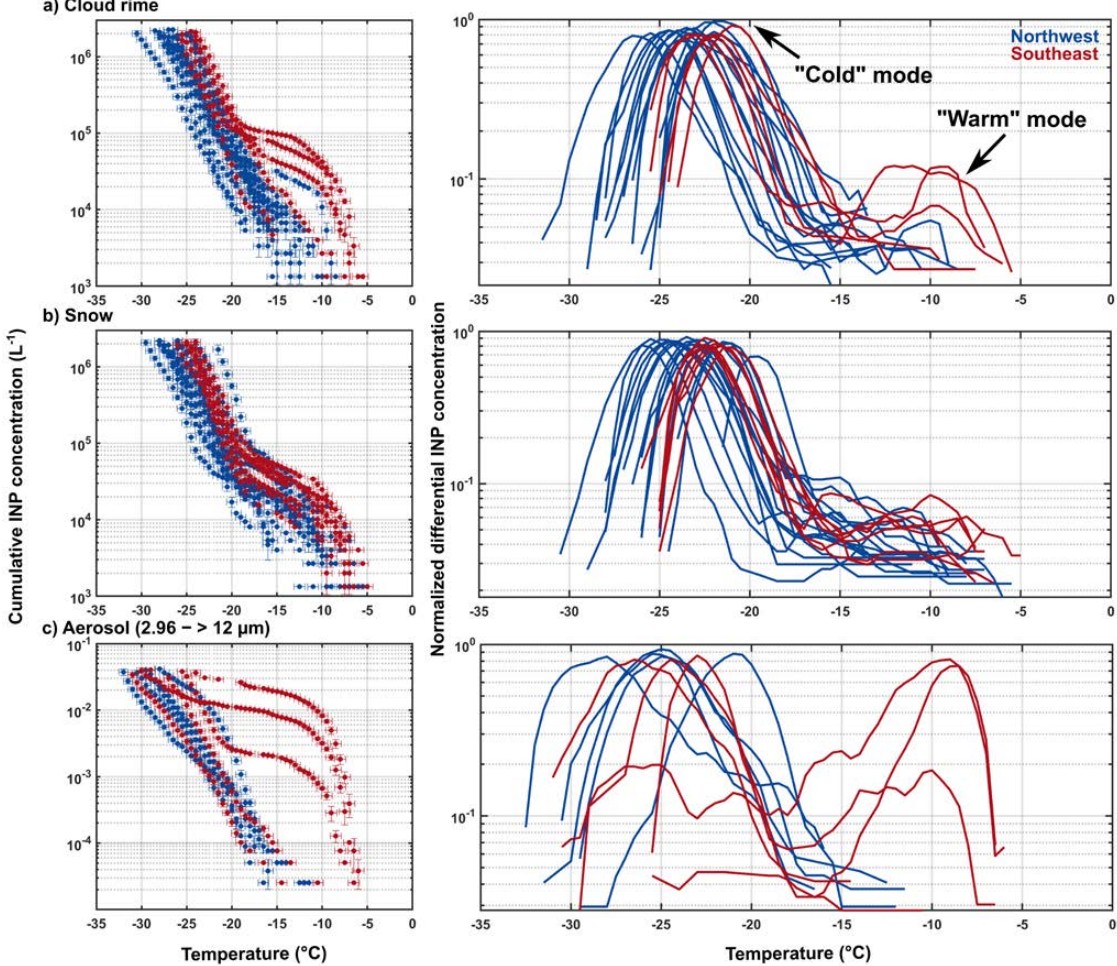

689

**Figure 7. Cumulative INP spectra (left) and normalize INP spectra (right) for the same samples of a) cloud rime, b) snow, and c) aerosol for the size range 2.96 − >12 μm in diameter. Spectra shown are for samples from the northwest (blue) and southeast (red) case study dates. Multiple cloud rime and snow samples were collected while one aerosol sample from each size range was collected on case study days (see Table 1). Additional dates with only aerosol samples (24-Feb and 27-Feb) are also shown in c) (highest of the two modes > −15 °C) and are discussed in section 3.3. The "cold" and "warm" modes are indicated in the normalize INP spectra for cloud rime, for reference.**



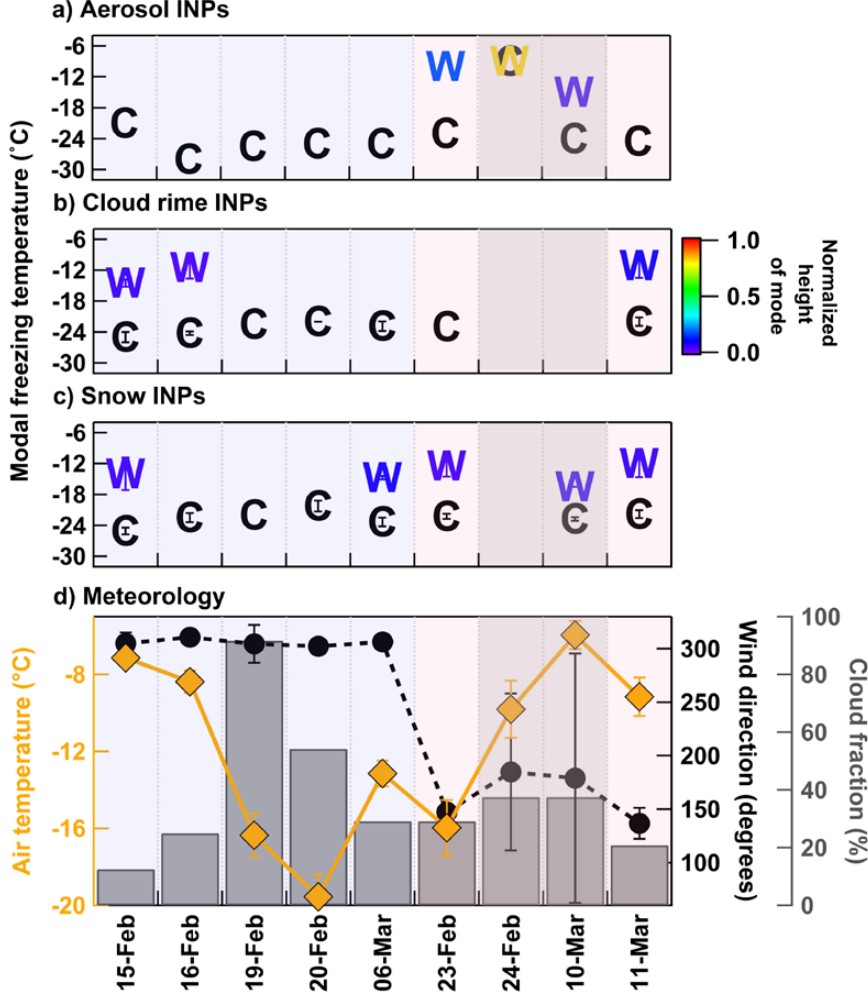

696

**Figure 8. Spectral statistics of cold mode height temperature and warm mode height temperatures denoted by "C" and "W", respectively for a) 2.96 − > 12 μm aerosol, b) cloud rime averaged per day, and c) snow averaged per day. Days with only "C" marker indicate the absence of a warm mode. d) shows average air temperature, wind direction, and cloud fraction during the case study days. The days are ordered by northwesterly (blue shading) and southeasterly (pink shading) case days. The southeasterly cases shaded in grey represent days that were not case study days, but days that help explain circumstances of the sampling on 23 Feb and 11 Mar.**