# Peer review of "Using freezing spectra characteristics to identify ice nucleating particle"

_Atmospheric Chemistry and Physics, 2018_

## Referee Comment (RC1) · Vali (Referee) · 14 Dec 2018

**Referee comments on "Using spectra characteristics to identify ice nucleating particle populations during winter storms in the Alps" by Creamean, Mignant, Bukowiecki and Conen.**

Gabor Vali

Department of Atmospheric Science, University of Wyoming, Laramie, Wyoming, USA

vali@uwyo.edu

**1   Summary**

The two related goals of the paper are: in-cloud measurements of INPs and the identification of INP type or composition. The paper makes a good contribution to INP studies by providing a data set involving INP measurements in air (with size sorting), rime, and snow at the same time. They show evidence for the dependence of INP abundance depending of air mass trajectory. Some inferences are drawn regarding local versus distant sources of INPs. The authors' conclusions are relatively simple and reasonably well supported by the data, but the analyses are limited without chemical and other supporting measurements. All that notwithstanding, an important revision is needed to take into account the points made in the Section 3 below.

**2   General points**

1) Stratification by storm type in Sections 3.1 and 3.2 focus only on air trajectory. It is surprising that precipitation during preceding days, cloud height, depth etc. are not considered. In some fashion those parameters are probably related to the airmass trajectories, but omission of attention to those factors makes the arguments presented sound incomplete and superficial. This sort of narrow focus in the treatment of the data is unsettling in various parts of the paper. Results in Section 3.2 and Fig. 6 are presented without separation of in-cloud and out-of-cloud samples. This sort of problem is not unique to this data set. It has also been there with the numerous papers reporting INP, and other, measurements at JFJ observatory.

2) The paper expands on other INP data from JFJ by determining INP abundance as a function of temperature, i.e. INP spectra. This is a useful step and avoids the somewhat simplistic comparisons that, unfortunately, arise from discussing and comparing INP concentrations without distinguishing between data obtained at $-10°$C or at $-30°$C. INP composition, their sources and their transport can be expected to be significantly different for widely different temperature regimes. The greater focus on this aspect of INP studies is a good contribution. Expanding further on the point, use of differential INP spectra for gaining more information about INPs is a commendable goal as it goes even further in focusing on INP characteristics than the more frequently employed cumulative spectra. However, the manner this is done in this paper deserves closer examination and revision, as argued under the heading ?Differential spectra? below.

3) As a general comment about interpretations of INP spectra, I'd express some caution. The paragraph in lines 267-285 about warm and cold-mode INPs is somewhat superficial. The temperature ranges and magnitudes of INP concentrations need to be considered quantitatively before deductions are made about possible sources of the INPs. Just looking at freezing temperatures (like onset and percentiles frozen) can be misleading since they depend on sample concentration (e.g. filter exposure time, dilution), number of drops tested and drop volume.

4) Related to the point raised in 1), one wonders what is underlying vision for comparing or correlating INP concentrations in air (via the filters) and in snow and in rime. These are three very different pathways for the INPs. Sources for each component can be different in space and time. With a broad separation such as NW versus SE airmasses perhaps these differences become unimportant but not necessarily so. It would be useful to read about the authors? perception on this issue. The paper glosses over such concerns thereby creating a degree of unease about the meaning of the results.

**3   Differential spectra**

5) What in this paper is called (Fig. 7, lines 26, 145, and other places) "normalized differential INP concentration" does not correspond to previously used definitions of that term. Earlier definitions are found in the references cited in the paper. Further references are Vali et al. (2015) (Atmos. Chem. Phys. 15, 10263-10270) and the recent Vali (2018) (Atmos. Meas. Tech. Discuss., https://doi.org/10.5194/amt-2018-309). To summarize, briefly, differential spectra, or differential INP concentrations, reflect the concentration of INPs or active sites per unit temperature interval. It also describes the probability of freezing for drops of given volume at given temperatures. As its name indicates, the differential spectrum corresponds to the differential of the cumulative spectra. The cumulative spectrum is denoted by $[INPs(T)]$ on line 147 of the paper. Other frequent designation of this cumulative spectrum[1] is $K(T)$, with $k(T)$ used for the differential spectrum. Both quantities are closely related to site density. Because the values are specific to each temperature, independently of what other INPs may be present in the sample, the differential spectra provide more acute diagnoses of INP characteristics then the more frequently employed cumulative spectra or the fraction frozen.

6) Line 152 states that differential values were obtained from the cumulative concentration. In fact, the results shown in Fig. 7 (right hand panels) and then used in Section 3.3 appear to represent some different quantity because differential spectra $k(T)$ usually exhibit a steady rise with decreasing temperatures at low temperatures. The large drop in the spectra at low temperatures seen in the right-hand panels of Fig. 7 suggest that what is shown in the graphs represent the differentials of the frequency of freezing events, $f(T)$, not of the cumulative INP concentration. Since fewer and fewer drops are left as the sample drops freeze and $f(T)$ approaches unity at the lowest temperatures in an experiment, the increase in $f(T)$ per temperature interval, $\delta f(T)/\delta T$, can be expected to drop off, just as it seen in these figures.

7) The differential $\delta f(T)/\delta T$ is a valid representation of the measurements but it lacks clear physical meaning. That differential is nearly the same as the 'freezing rate, R(T)' defined in Section 4.6 of Vali et al. (2015), but without the first ratio in
* * *
[1]Symbols used in here follow Vali et al. (2015)

that definition. For $f(T) \ll 1$, i.e. at the higher (warm) end of the range $\delta f(T)/\delta T$ the first term in $R(T)$ is close to unity and $\delta f(T)/\delta T \simeq R(T)$ so that minor peaks are resolved in $\delta f(T)/\delta T$ without the distorting effect of reduced sample size. However, at smaller values of $f(T)$ that distortion becomes dominant, so that little significance can be attached to peaks such as those shown in Fig. 7 of the paper. That this is indeed the case can be demonstrated by changing the drop volumes. Such a

5    change would shift the positions of the cold-mode peak. In contrast, $k(T)$ for different drop sizes overlap and form a continuous curve (e.g. Fig. 4 in Vali (1971)(J. Atmos. Sci., 28,402). Another illustration can be thought of with dilutions of the sample with INP-free water. The drop-off beyond the cold-mode peak of the original sample will not be reproduced with the diluted sample since large numbers of drops will be still available for substantial rates of freezing at that temperature. In this case too, $k(T)$ for successive dilutions overlap and form a continuous curve.

10    The ratio $\delta f(T)/\delta T$ has not been used in the past to describe experimental results, so the authors should define it clearly and explain why they chose that presentation. Renaming that quantity would be helpful in avoiding future misunderstandings. Alternatively, they could consider changing the analysis to using $k(T)$ or $R(T)$ to represent their data. The point is that $\delta f(T)/\delta T$ is an experiment-specific relative characterization of the results, while other, more objective quantities could be used.

15    An additional effect of the choice of analysis in terms of $\delta f(T)/\delta T$ is that freezing of some of the the drops at warmer temperatures decreases the number and, potentially, the gradient of the frequency at lower temperatures. Is that the reason why the so-called cold-mode for the SE samples seems to occur at warmer temperatures in Fig. 7 than for the NW samples?

**4   Specific points**

lines 33-34:    Precipitation production is not limited, as implied, to clouds that contain both liquid and ice, Unfortunate
20          phrasing. But the mention of mixed-phase clouds (MPC) in the subsequent several lines keep mixing the focus on the importance of INPs in general and the importance of MPCs for precipitation production.

line 37:    There is jump here from the discussion of the INPs as important problems for weather and climate models to the practical problem of measuring INPs. Also, from none of the foregoing follows the argument of line 41 that in-cloud measurements are indispensable fo progress.

25   line 43:    Another jump in logic. What?s said here had to be already taken for granted for the previous paragraph to make sense.

lines 43-76:    These two paragraphs get into too much detail. The present study does not address many of the details described as possibly controversial or uncertain, so raising the issues is a diversion. Other review papers deal with what is known about the components of INPs. This paper is directed only to a broad classification of INP
30          types by composition.

lines 104-105:    To what extent did wind removal of snow from the pans hinder determinations of snowfall rates?

line 131:    Drop size variability introduces errors due to INP content being proportional to volume, as also pointed out in Creamean et al. 2018b. How significant this error is depends on the range of variation of drop volumes. The way

this line is phrased is incorrect and subjective. The error introduced may be small or equal compared to other sources. The only concise way to test for this would be to do large numbers of repeated runs from the same sample. Recommend to change ?indeterminable? to ?undetermined? in line 130 and eliminate line 131.

line 136:      Cooling-rate dependence is small but not non-existent as shown in the references cited. If the authors? tests showed no discernible dependency it must be because other variations hid the cooling-rate dependence.

line 142:      The reader deserves to know what this 'custom software' was that connects visual detection with a recording system. Also, one wonders why is cooling rate considered a factor for each drop, when the previous paragraph states that there is no 'discernible' effect.

line 144:      How were triplicate samples combined for analyses? Averages of fraction frozen at each temperature? How much variability was there among the three runs per sample?

line 153:      Assigning significance to the fact that the data shown in the references cited extended to only about -20°C may indicate a misunderstanding. The -20°C limit was due to no fundamental limitation of the validity of definition of differential spectra. It was due to the background from distilled water and from the supporting surface becoming important at colder temperatures.

lines 160-163:      Can the method used distinguish between clouds enveloping the observatory and clouds just a few hundred meters above it? Were there no in-situ instruments or visibility measurements available for detection of clouds?

line 173:      Please define SDE.

Fig. 6 panel a:      Are the lines shown averages for the each condition? If the curves in panel a) are differential concentrations, as the presence of a peak suggests, then the units of the ordinate are incorrect.

Fig. 6 panel b-h:      The numbers of points doesn't seem to be the same in each panel. Some points may be missing for samples that had no freezing event at -10°C, but shouldn't the other panels contain the same numbers of points. It is hard to tell if that is the case. Perhaps due to overlap of points. Please check and explain if some additional selection has been made.

line 233->:      It would be helpful if the authors stated what signals they consider significant in Fig. 6. The figure is complex and the data noisy. The jump into comparisons with the Stopelli data is hard to follow without first pointing out what deductions are extracted from Fig. 6.

lines 239-240:      The phrasing could be improved. You mean cumulative concentrations at -15°C as the criterion used for the assertion?

line 243:      Onset definition is based on single drops? Wording in parenthesis on lines 244-245 should be corrected and made more specific.

lines 265-266:      This assertion about the role of hoar frost needs some explanation.

lines 267-285:      This paragraph will need to be revisited after the authors examine the issue raised about the differntial spectra, and also take into account the comment on line 153 above.

Section 3.3:      Because of misgivings about the meaning of the cold-mode data, it is difficult to sort out what part of the conclusions is supported by the data. Focusing only on whether a minor peak was present or not and forgetting

about the cold-mode $\mathbf{W}$ entries in Fig. 8 the analysis may not change much since the position of the $\mathbf{C}$ peaks is almost uniform for all the days and contains little additional information.

line 348:    Reference to 'differential INP spectra' need change in light of Section 3 of these comments.

line 351:    The meaning of statements about the relative magnitudes of INP concentrations is unclear unless the temperature to which they refer is specified, and it is clear what quantity is used for the comparison ($f(T)$ or $k(T)$ or $K(T)$, or other).

line 352:    Reference to bi-modal spectra may also need to be re-thought depending on what changes are made regarding the use of $\delta f(T)/\delta T$) or its replacement. The

line 354-359:    The thoughts expressed here are basically sound but are somewhat overstated regarding how much information can be gained from cumulative versus differential spectra. The latter are more specific and make it easier to see differences among samples, but the information content is not different.

---

## Referee Comment (RC2) · Anonymous Referee #2 · 14 Dec 2018

Summary:

The results presented by M. Creamean and co-authors give insights into potential sources of INPs at the High Altitude Research Station Jungfraujoch. The study was conducted in winter and investigates freezing spectra of aerosol particles as derived from aerosol, cloud rime, and snow samples. The results from this study give an interesting insight into INP characteristics during this specific time at Jungfraujoch, however, my concerns center mostly around the methods used, which impact the interpretation of the results. After addressing these concerns and better highlighting the limitations of the study, the manuscript will be suitable for publication in ACP.

[Figure]

Major remarks:

- The results presented by Creamean et al. are based on a limited set of data with respect to sampling time. E.g. none of the southeasterly samples are within the free troposphere. This results in a confusion of air mass characteristics by wind direction versus in-or-out of boundary layer conditions. As such I believe the air masses cannot be distinguished as function of wind direction, which is one of the main findings of the authors. I suggest to better indicate which air mass is transported in the free troposphere and which is impacted by the boundary layer.

- Jungfraujoch is regularly exposed to short-term emissions from touristic activities, as e.g. tobacco smoke, emissions from helicopters and snow cats, as summarized in Bukowiecki et al., 2016. Such local emissions can lead to short-term peak concentrations of aerosol particles. Did you consider such emissions, especially with regard to the 24-hr aerosol sample? If not, this limits the interpretation of the results and should be pointed out.

- Tracing air mass origin with the HYSPLIT model has some limitations in a complex terrain as Jungfraujoch. Another powerful tool to determine source emission sensitivities is FLEXPART, which is specifically improuved for the site (e.g. Pandey Deolal et al., 2014). Given the spatial resolution of HYSPLIT I find it hard to thrust interpretations based on such results alone.

- To my understanding the collection of cloud rime should only result in impaction of liquid water droplets on the collection plate, which freeze upon collision. In case that the cloud temperature is colder than the activation temperature of INPs, such samples should not contain ice-active particles. However, INP concentrations are very high in rime samples, and orders of magnitudes higher than the aerosol sample INP concentration. Please explain why this is the case.

Specific remarks:

Page 1, title: Please specify that you investigate freezing spectra characteristics; readers outside of the INP community might get confused.

Page 1, lines 26 – 28 (and page 8, lines 284 – 285): I believe that with the methods used here, you can not clearly identify biological, dust, or a mixture between the two. To make such connection in field studies would require a proper assessment of the aerosol particle population and/or of the ice residuals with respect to the biological and dust particle concentration.

Page 1, line 30: This statement needs a reference.

Page 1, lines 33 – 34: The statement on the microphysical impact on precipitation formation in mixed-phase clouds needs a reference.

Page 3, lines 86 – 87: Before you defined "warm temperature INPs" as INPs active > -15°C (page 2, line 72). Please be consistent.

Page 3, lines 97 – 98: Given the different temporal and spatial resolutions of your aerosol, cloud rime and snow samples, how can you explain the exchange of INPs into air, cloud, and precipitation? E.g. a 24-hr aerosol sample might not be dominated by the INP population which caused cloud and precipitation formation. Also, clouds might from far away from the site, as well as precipitation particles before reaching the ground.

Page 3, line 111: Please indicate if you refer to volume or mass/standard flow.

Page 4, line 116: What is the collection efficiency of the drums? Is there a size dependency, e.g. an increased loss due to reflection on the stages for larger particles?

Page 5, line 164: You introduce 222Rn as abbreviation for radon, but you do not use it consistently in the text. Or do I miss a major difference between "radon" and "222Rn"?

Page 5, line 177: You do not use the abbreviated "TSP" thereafter, therefore I suggest to not introduce it.

Page 5, line 177: I do not under stand why an increase in the 48-hour total suspended particle concentration is of interest in this study; given the time resolution of your rime and snow sample is on the order of hours, an overlap between a detected SDE (> 4 hours) and your sample might have occurred.

Page 5, lines 168 – 169: In previous studies several approaches have been used to assess boundary layer contact of the air mass arriving at Jungfraujoch. To make the distinction between boundary layer influence and free tropospheric conditions more reliable I suggest to use an additional method, as e.g. described by Herrmann et al., 2015.

Page 6, lines 200 – 201: This is very vague, and not well quantified. A statistical analysis of e.g. mean travel height of the back trajectories for both wind directions would be helpful.

Page 6, line 220: What is the correlation coefficient? Is the relationship statistically significant?

Page 7, line 221: How do you determine upslope winds out of figure 1b?

Page 7, lines 236 – 237: Please add another reference for this quite general statement.

Page 7, line 252 – 257: Figures 6c-h are hard to understand in the way they are visualized. Given that you do not observe any relationship between INP concentration and air temperature/wind speed, I suggest to show these results in in the supplementary material.

Page 9, line 300: I cannot identify the freezing spectra for this case study. Please highlight this in e.g. figure 7.

Page 9, lines 300 – 302: I find it hard to follow your argumentation since you do not know the source region and transport pathway of the cloud.

Page 9, lines 331 – 333: In order to strengthen this finding, you could include meteorological maps indicating frontal systems.

Page 10, lines 354 – 368: This main part of your conclusion is rather open discussion, perspective and recommendation. I do not see relevant conclusions based on the presented results in this section.

Page 18, figure 1: Apparently, the rime and snow collection time was often longer than the actual occurrence of the cloud event, since the grey shading in figure 1a indicated that the cloud events were lasting shorter than the sampling period (figure 1b). If so, please specify in the text.

Page 18, figure 1: The labels of the y-axis should not only contain the units, but also the property (e.g. "relative humidity (%)")

---

## Author Comment (AC1) · 27 Feb 2019

*We would like to thank the reviewers for their insightful commentary and catching the inconsistencies that existed in the first version. As a result, we have revised the manuscript substantially and believe it has significantly improved from its original submission. Please see the tracked changes revision attached to this review. One change that we made on our own volition was to develop a separate sub-category of the southeasterly cases that were either completely or predominantly southeasterly, and were either SDE days (i.e., 24 Feb and 10 Mar) or days with the most boundary layer influence based on the radon data (i.e., 27 Feb and 28 Feb). These days are indicated in green in the table, and figures and discussion has been revised to reflect this change.*

**Summary:**

The two related goals of the paper are: in-cloud measurements of INPs and the identification of INP type or composition. The paper makes a good contribution to INP studies by providing a data set involving INP measurements in air (with size sorting), rime, and snow at the same time. They show evidence for the dependence of INP abundance depending of air mass trajectory. Some inferences are drawn regarding local versus distant sources of INPs. The authors' conclusions are relatively simple and reasonably well supported by the data, but the analyses are limited without chemical and other supporting measurements. All that notwithstanding, an important revision is needed to take into account the points made in the Section 3 below.

**Major remarks:**

1) Stratification by storm type in Sections 3.1 and 3.2 focus only on air trajectory. It is surprising that precipitation during 10 preceding days, cloud height, depth etc. are not considered. In some fashion those parameters are probably related to the airmass trajectories, but omission of attention to those factors makes the arguments presented sound incomplete and superficial. This sort of narrow focus in the treatment of the data is unsettling in various parts of the paper. Results in Section 3.2 and Fig. 6 are presented without separation of in-cloud and out-of-cloud samples. This sort of problem is not unique to this data set. It has also been there with the numerous papers reporting INP, and other, measurements at JFJ observatory.

*The intention of the classification was to simply determine directionality of the storms—and thus, possible INP sources—affecting Jungfraujoch during the study as delineated by section 3.1. The two dominant flow patterns are well documented at Jungfraujoch (i.e., Stopelli et al. (2015) and the newly-added reference of Collaud Coen et al. (2011)). In the current work, we use air mass trajectories, but also corroborate with local wind speed and direction, and evaluate INPs in the context of air temperature during storm days in section 3.2.*

*We do not classify by "storm type" since adequate parameters such as precipitation quantities and cloud properties were not measured and are not routine baseline measurements at Jungfraujoch. For clarity, we now provide a definition of "storms" in the beginning of section 3.1: "These days were also deemed days with "storm" conditions since clouds and snow were both present at JFJ." We also now define that we evaluate INPs in the context of storm directionality where appropriate throughout the revision (e.g., in section 3.2). However, we have included some non-storm days in the revision and initially define those in section 3.1, so we have reduced the use of "storm" cases unless only referring to the northwest or southeast samples.*

*We already presented cloud percentage data in Figures 1 and 8 (now presented in the new Figure 1 only) using the methodology of Herrmann et al. (2015), which show that all case study samples were collected under at least some percentage of in-cloud conditions. This agrees with the fact that cloud rime was collected during our case study days, indicating the presence of clouds. Presenting the cloud data a third time in Figure 6 would be redundant.*

Collaud Coen, M., Weingartner, E., Furger, M., Nyeki, S., Prevot, A. S. H., Steinbacher, M., and Baltensperger, U.: Aerosol climatology and planetary boundary influence at the Jungfraujoch analyzed by synoptic weather types, Atmos Chem Phys, 11, 5931-5944, 10.5194/acp-11-5931-2011, 2011.

2) The paper expands on other INP data from JFJ by determining INP abundance as a function of temperature, i.e. INP spectra. This is a useful step and avoids the somewhat simplistic comparisons that, unfortunately, arise from discussing and comparing INP concentrations without distinguishing between data obtained at -10°C or at -30°C. INP composition, their sources and their transport can be expected to be significantly different for widely different temperature regimes. The greater focus on this aspect of INP studies is a good contribution. Expanding further on the point, use of differential INP spectra for gaining more information about INPs is a commendable goal as it goes even further in focusing on INP characteristics than the more frequently employed cumulative spectra. However, the manner this is done in this paper deserves closer examination and revision, as argued under the heading "Differential spectra" below.

*We have revised the differential spectral calculations, figures, and discussion significantly to follow the methodology of Vali (2018). Please see detailed responses in the "Differential spectra" section below.*

3) As a general comment about interpretations of INP spectra, I'd express some caution. The paragraph in lines 267-285 about warm and cold-mode INPs is somewhat superficial. The temperature ranges and magnitudes of INP concentrations need to be considered quantitatively before deductions are made about possible sources of the INPs. Just looking at freezing temperatures (like onset and percentiles frozen) can be misleading since they depend on sample concentration (e.g. filter exposure time, dilution), number of drops tested and drop volume.

*We now chiefly focus on the differential spectra but still do mention the cold and warm modes in the original df/dT spectra, which are now correctly labeled. We have found df/dT useful in qualitative evaluation of possible different INP populations (i.e., what are likely modes for mineral + biological versus biological) within one sample, as demonstrated additionally in previous work by Augustin et al. (2013). Almost all of the samples (except for the 24-Feb and 28-Feb aerosol samples) contain spectra that have a cold mode when assessing df/dT, but we would like to show their temperatures to demonstrate the dT between each of the two modes when a warm mode is present. The warm mode in the differential spectra afford a more quantifiable approach of the "bump" in the warm regime of the cumulative spectra as observed in previous studies (e.g., DeMott et al., 2016; DeMott et al., 2018; Hill et al., 2016; Kanji et al., 2017; McCluskey et al., 2017; Petters and Wright, 2015; Suski et al., 2018; Vali, 1971; Vali and Stansbury, 1966).*

*Hence, we decided to show both spectra types in addition to cumulative and have clarified throughout the text which we refer to. We also have added explanation and discussion on all three types of spectra and why they are useful in section 3.2. The new Figure 5 contains all three spectra types, but Figure 6 shows mode temperatures for df/dT since the differential spectra have the same values for the warm mode (as we now note in the caption and discussion) and no mode in the cold regime due to exponential increase at those temperatures. We use this combination of spectral information to glean possible sources.*

Augustin, S., Wex, H., Niedermeier, D., Pummer, B., Grothe, H., Hartmann, S., Tomsche, L., Clauss, T., Voigtländer, J., Ignatius, K., and Stratmann, F.: Immersion freezing of birch pollen washing water, Atmos. Chem. Phys., 13, 10989-11003, 10.5194/acp-13-10989-2013, 2013.

4) Related to the point raised in 1), one wonders what is underlying vision for comparing or correlating INP concentrations in air (via the filters) and in snow and in rime. These are three very different pathways for the INPs. Sources for each component can be different in space and time. With a broad separation such as NW versus SE airmasses perhaps these differences become unimportant but not necessarily so. It would be useful to read about the authors' perception on this issue. The paper glosses over such concerns thereby creating a degree of unease about the meaning of the results.

*The difference in temporal coverage between snow, rime, and aerosol sampling was due to practical constraints. It forces us to integrate in our interpretation over an air mass including clouds and cloud-free sections. As long as wind direction and planetary boundary layer influence are similar within an air mass, we think this approach can still lead to insights, although a perfect synchrony in sampling all components would of course have been preferable. We now address this issue at the end of section 2.1 (Aerosol, cloud rime and snow collection at Jungfraujoch).*

**Differential spectra:**

5) What in this paper is called (Fig. 7, lines 26, 145, and other places) "normalized differential INP concentration" does not correspond to previously used definitions of that term. Earlier definitions are found in the references cited in the paper. Further references are Vali et al. (2015) (Atmos. Chem. Phys. 15, 10263-10270) and the recent Vali (2018) (Atmos. Meas. Tech. 15 Discuss., https://doi.org/10.5194/amt-2018-309). To summarize, briefly, differential spectra, or differential INP concentrations, reflect the concentration of INPs or active sites per unit temperature interval. It also describes the probability of freezing for drops of given volume at given temperatures. As its name indicates, the differential spectrum corresponds to the differential of the cumulative spectra. The cumulative spectrum is denoted by [INPs(T)] on line 147 of the paper. Other frequent designation of this cumulative spectrum is K(T), with k(T) used for the differential spectrum. Both quantities are closely related to site density. Because the values are specific to each temperature, independently of what other INPs may be present in the sample, the differential spectra provide more acute diagnoses of INP characteristics then the more frequently employed cumulative spectra or the fraction frozen.

*We have revised the terminology to be consistent with Vali (2015, 2018). Specifically, we redefined cumulative INP concentrations or what we called [INPs(T)] as K(T) and differential INP concentrations as k(T). We relabeled axes in the figures and updated the text to be consistent with these definitions, and added more details on the calculations used in section 2.2 for clarity.*

6) Line 152 states that differential values were obtained from the cumulative concentration. In fact, the results shown in Fig. 7 (right hand panels) and then used in Section 3.3 appear to represent some different quantity because differential spectra k(T) usually exhibit a steady rise with decreasing temperatures at low temperatures. The large drop in the spectra at low temperatures seen in the right-hand panels of Fig. 7 suggest that what is shown in the graphs represent the differentials of the frequency of freezing events, f(T), not of the cumulative INP concentration. Since fewer and fewer drops are left as the sample drops freeze and f(T) approaches unity at the lowest temperatures in an experiment, the increase in f(T) per temperature interval, δf(T)/δT, can be expected to drop off, just as it seen in these figures.

*We realize we had previously only shown the differential of f(T) (i.e., δf(T)/δT or df/dT as we call it to be consistent with the notation presented by Augustin et al. (2013))—thank you for catching this inconsistency. We redid all calculations and now provide the k(T) equation from Vali (2018) that we applied to our data (see section 2.2). We now show K(T), df/dT, and k(T) in the new Figure 5.*

*Bimodality was evident in the df/dT spectra, however, given the large increase in k(T) in the cold temperature (i.e., < −15 °C) regime, well-defined "cold modes" are not present. We now clarify that when we refer to cold modes, that is for df/dT only and warm modes are for both df/dT and k(T). We have highlighted these regions in each of the spectra types in Figure 5.*

7) The differential δf(T)/δT is a valid representation of the measurements but it lacks clear physical meaning. That differential is nearly the same as the 'freezing rate, R(T)' defined in Section 4.6 of Vali et al. (2015), but without the first ratio in that definition. For f(T) << 1, i.e. at the higher (warm) end of the range δf(T)/δT the first term in R(T) is close to unity and δf(T)/δT ~ R(T) so that minor peaks are resolved in δf(T)/δT without the distorting effect of reduced sample size. However, at smaller values of f(T) that distortion becomes dominant, so that little significance can be attached to peaks such as those shown in Fig. 7 of the paper. That this is indeed the case can be demonstrated by changing the drop volumes. Such a change would shift the positions of the cold-mode peak. In contrast, k(T) for different drop sizes overlap and form a continuous curve (e.g. Fig. 4 in Vali (1971)(J. Atmos. Sci., 28,402). Another illustration can be thought of with dilutions of the sample with INP-free water. The drop-off beyond the cold-mode peak of the original sample will not be reproduced with the diluted sample since large numbers of drops will be still available for substantial rates of freezing at that temperature. In this case too, k(T) for successive dilutions overlap and form a continuous curve. The ratio δf(T)/δT has not been used in the past to describe experimental results, so the authors should define it clearly and explain why they chose that presentation. Renaming that quantity would be helpful in avoiding future misunderstandings. Alternatively, they could consider changing the analysis to using k(T) or R(T) to represent their data. The point is that δf(T)/δT is an experiment-specific relative characterization of the results, while other, more objective quantities could be used. An additional effect of the choice of analysis in terms of δf(T)/δT is that freezing of some of the drops at warmer temperatures decreases the number and, potentially, the gradient of the frequency at lower temperatures. Is that the reason why the so-called cold-mode for the SE samples seems to occur at warmer temperatures in Fig. 7 than for the NW samples?

*We have found df/dT useful in qualitative evaluation of possible different INP populations (i.e., what are likely modes for mineral + biological versus biological) within one sample, as demonstrated additionally in previous work by Augustin et al. (2013). We realize this may be experiment-specific and make note of this in section 3.2 – that the use of df/dT is to intercompare samples in this study. We are now careful with discussion on these spectra and are transparent by explaining their qualitative nature of showing either one or two of the INP populations.*

**Specific remarks:**

lines 33-34: Precipitation production is not limited, as implied, to clouds that contain both liquid and ice, Unfortunate phrasing. But the mention of mixed-phase clouds (MPC) in the subsequent several lines keep mixing the focus on the importance of INPs in general and the importance of MPCs for precipitation production.

*For clarity, we have changed this sentence to, "Aerosol-induced ice microphysical modifications influence cloud lifetime and albedo (Albrecht, 1989; Twomey, 1977; Storelvmo et al., 2011), as well as the production of precipitation (DeMott et al., 2010)."*

line 37: There is jump here from the discussion of the INPs as important problems for weather and climate models to the practical problem of measuring INPs. Also, from none of the foregoing follows the argument of line 41 that in-cloud measurements are indispensable for progress.

*We moved around a couple of sentences in this paragraph to start with only broader aerosol-cloud statements and then transition to INPs. We also changed "necessary" to "useful for assessing" to omit any idea that these are merely indispensable for progress.*

line 43: Another jump in logic. What's said here had to be already taken for granted for the previous paragraph to make sense.

*Because we reorganized this paragraph, we left the last sentence as it demonstrates that INP observations in-cloud are even more limited than observations in general (i.e., in both in-cloud and below-cloud or ground level measurements when measurements are not conducted at levels where clouds can form). Perhaps there is some confusion with "in cloudy" compared to "in-cloud". We have now clarified we are referring to directly in-cloud.*

lines 43-76: These two paragraphs get into too much detail. The present study does not address many of the details described as possibly controversial or uncertain, so raising the issues is a diversion. Other review papers deal with what is known about the components of INPs. This paper is directed only to a broad classification of INP types by composition.

*To provide context for the spectral characteristics and what they might mean, we need such detailed background on what types of INPs call where on the spectrum of freezing temperatures. However, we realize we cannot address the limitations that we call out from previous studies, so we remove mentioning of those uncertainties or controversial nature. We retain the limitations of the modeling examples, as they support the need for additional observations (not just ours, but in general, more are needed to improve the models).*

lines 104-105: To what extent did wind removal of snow from the pans hinder determinations of snowfall rates?

*Unfortunately, we can only speculate that winds did remove some of the snow but cannot quantitatively assess this to provide actual snow rates. We note wind removal is a possibility and thus cannot determine snowfall rates in the beginning of section 2.1.*

line 131: Drop size variability introduces errors due to INP content being proportional to volume, as also pointed out in Creamean et al. 2018b. How significant this error is depends on the range of variation of drop volumes. The way this line is phrased is incorrect and subjective. The error introduced may be small or equal compared to other sources. The only concise way to test for this would be to do large numbers of repeated runs from the same sample. Recommend to change "indeterminable" to "undetermined" in line 130 and eliminate line 131.

*Done.*

line 136: Cooling-rate dependence is small but not non-existent as shown in the references cited. If the authors' tests showed no discernible dependency it must be because other variations hid the cooling-rate dependence.

*Good point. We have changed "no discernible" to "very little discernible".*

line 142: The reader deserves to know what this 'custom software' was that connects visual detection with a recording system. Also, one wonders why is cooling rate considered a factor for each drop, when the previous paragraph states that there is no 'discernible' effect.

*It is simply software that records the time, probe temperature, and cooling rate every second. When we identify that a drop has frozen, we click a button so that the software records that exact time, probe temperature, and cooling rate of that drop in a separate file. We now provide this level of detail here.*

line 144: How were triplicate samples combined for analyses? Averages of fraction frozen at each temperature? How much variability was there among the three runs per sample?

*We have defined in our previous publications that the 3 tests typically do not vary drastically from each other, and any variability is considered when calculating the error bars. We first combine the frozen drop records from the three tests, then calculate fraction frozen, then INP concentrations. We have clarified this in section 2.2.*

line 153: Assigning significance to the fact that the data shown in the references cited extended to only about -20°C may indicate a misunderstanding. The -20°C limit was due to no fundamental limitation of the validity of definition of differential spectra. It was due to the background from distilled water and from the supporting surface becoming important at colder temperatures.

*Thank you for pointing this out. We have changed this sentence to, "Spectra from these previous studies only reached a minimum of –20 ˚C due to the limitations of background artifacts in the water used at that time."*

lines 160-163: Can the method used distinguish between clouds enveloping the observatory and clouds just a few hundred meters above it? Were there no in-situ instruments or visibility measurements available for detection of clouds?

*Unfortunately, there were no in situ cloud or visibility measurements. We added a statement delineating this missing element in the beginning of section 2.3. The effective sky temperature method we used from Hermann et al. (2015) demonstrates robustness in the winter at Jungfraujoch when compare to all-sky camera observations of cloud (i.e., both the method and all sky determine Jungfraujoch is in cloud 38% of the time in the winter), indicating the reliability of this method especially during this season. However, we cannot distinguish between clouds enveloping at the observatory and clouds a few hundred meters above. Given the scale of the storms hitting the site during INCAS (as evaluated from MODIS visible imagery; see representative examples from 15 Feb (top) and 24 Feb (bottom) below from https://worldview.earthdata.nasa.gov/), it is likely that the localized orographic formation is rare in comparison, rendering the issue irrelevant.*

[Figure]

line 173: Please define SDE.

*SDE was already defined in the introduction.*

Fig. 6 panel a: Are the lines shown averages for the each condition? If the curves in panel a) are differential concentrations, as the presence of a peak suggests, then the units of the ordinate are incorrect.

*These were averages of cumulative spectra. We originally did not include zero as INP values before the onset freezing temperatures, which is why it did not look correct or have a cumulative shape. We have fixed this and now label the plots to define when they are cumulative or otherwise.*

Fig. 6 panel b-h: The numbers of points doesn't seem to be the same in each panel. Some points may be missing for samples that had no freezing event at -10°C, but shouldn't the other panels contain the same numbers of points. It is hard to tell if that is the case. Perhaps due to overlap of points. Please check and explain if some additional selection has been made.

*This is due to overlap of some points, which we now state in the caption: "Some data points overlap and thus plots may appear to not have the same number of points per sample."*

line 233->: It would be helpful if the authors stated what signals they consider significant in Fig. 6. The figure is complex and the data noisy. The jump into comparisons with the Stopelli data is hard to follow without first pointing out what deductions are extracted from Fig. 6.

*We have significantly changed Figure 6 to make it less complex and noisy and now discuss it after showing the three different types of spectra. We segue from Figure 5 (spectra) to Figure 6 (statistics) more smoothly now so that the comparison with Stopelli et al. (2016) data make sense.*

lines 239-240: The phrasing could be improved. You mean cumulative concentrations at -15°C as the criterion used for the assertion?

*We did mean cumulative and now define this. Anytime we mention "INP concentrations", we now define it we are talking about cumulative or differential.*

line 243: Onset definition is based on single drops? Wording in parenthesis on lines 244-245 should be corrected and made more specific.

*Yes. We have no clarified this.*

lines 265-266: This assertion about the role of hoar frost needs some explanation.

*We have added to this sentence that hoar frost is a form of rime. Therefore, any addition of hoar frost to a collected snow sample will make it more similar to rime.*

lines 267-285: This paragraph will need to be revisited after the authors examine the issue raised about the differential spectra, and also take into account the comment on line 153 above.

*Done. This entire section was revised based on the addition of the correct differential spectra.*

Section 3.3: Because of misgivings about the meaning of the cold-mode data, it is difficult to sort out what part of the conclusions is supported by the data. Focusing only on whether a minor peak was present or not and forgetting about the cold-mode W entries in Fig. 8 the analysis may not change much since the position of the C peaks is almost uniform for all the days and contains little additional information.

*We have significantly revised this section to better explain what the results show by using a combination of cumulative, df/dT, and differential spectra.*

line 348: Reference to 'differential INP spectra' need change in light of Section 3 of these comments.

*We have fixed this to include the correct deductions based on the df/dT and correct differential spectra.*

line 351: The meaning of statements about the relative magnitudes of INP concentrations is unclear unless the temperature to which they refer is specified, and it is clear what quantity is used for the comparison (f(T) or k(T) or K(T), or other).

*We have defined the INP type, here and elsewhere in the manuscript for any mention of INP concentrations.*

line 352: Reference to bi-modal spectra may also need to be re-thought depending on what changes are made regarding the use of δf(T)= δT) or its replacement. The line 354-359: The thoughts expressed here are basically sound but are somewhat overstated regarding how much information can be gained from cumulative versus differential spectra. The latter are more specific and make it easier to see differences among samples, but the information content is not different.

*Based on the new calculations, our conclusions have been revised to properly fit what the data show.*

[revised manuscript text omitted]

---

## Author Comment (AC2) · 27 Feb 2019

*We would like to thank the reviewers for their insightful commentary and catching the inconsistencies that existed in the first version. As a result, we have revised the manuscript substantially and believe it has significantly improved from its original submission. Please see the tracked changes revision attached to this review. One change that we made on our own volition was to develop a separate sub-category of the southeasterly cases that were either completely or predominantly southeasterly, and were either SDE days (i.e., 24 Feb and 10 Mar) or days with the most boundary layer influence based on the radon data (i.e., 27 Feb and 28 Feb). These days are indicated in green in the table, and figures and discussion has been revised to reflect this change.*

**Summary:**

The results presented by M. Creamean and co-authors give insights into potential sources of INPs at the High Altitude Research Station Jungfraujoch. The study was conducted in winter and investigates freezing spectra of aerosol particles as derived from aerosol, cloud rime, and snow samples. The results from this study give an interesting insight into INP characteristics during this specific time at Jungfraujoch, however, my concerns center mostly around the methods used, which impact the interpretation of the results. After addressing these concerns and better highlighting the limitations of the study, the manuscript will be suitable for publication in ACP.

**Major remarks:**

- The results presented by Creamean et al. are based on a limited set of data with respect to sampling time. E.g. none of the southeasterly samples are within the free troposphere. This results in a confusion of air mass characteristics by wind direction versus in-or-out of boundary layer conditions. As such I believe the air masses cannot be distinguished as function of wind direction, which is one of the main findings of the authors. I suggest to better indicate which air mass is transported in the free troposphere and which is impacted by the boundary layer.

*Actually, the southeasterly sample from 24 Feb was in the free troposphere as demonstrated by the radon data in the new Figure 1. To avoid confusion and as the reviewer suggests, we now clarify which cases (with the newly added boundary layer days of 27 Feb and 28 Feb) were affected by boundary layer air and which by the free troposphere. Table 1 now includes classifications for days under predominantly free tropospheric or boundary layer influences.*

- Jungfraujoch is regularly exposed to short-term emissions from touristic activities, as e.g. tobacco smoke, emissions from helicopters and snow cats, as summarized in Bukowiecki et al., 2016. Such local emissions can lead to short-term peak concentrations of aerosol particles. Did you consider such emissions, especially with regard to the 24-hr aerosol sample? If not, this limits the interpretation of the results and should be pointed out.

*Although it is possible such sources could affect the particle population, it is unlikely the 3 – 12 µm particles were affected by such local sources of pollution, given the typical size distributions of fresh cigarette smoke and combustion aerosol (e.g., Frohlich et al. (2015); Li and Hopke (1993); Zhang et al. (2013)). Bukowiecki et al. (2016) discuss these local source possibilities in the context of carbonaceous aerosol in PM$_1$. Although this is the case, we did add a statement to section 2.1: "It is possible local sources of aerosol, such as tobacco smoke or emissions from touristic infrastructure were collected by the DRUM (Bukowiecki et al., 2016), but did not likely affect the 2.96 – >12 µm particles which we focus on herein.*

*Fröhlich, R., Cubison, M. J., Slowik, J. G., Bukowiecki, N., Canonaco, F., Croteau, P. L., Gysel, M., Henne, S., Herrmann, E., Jayne, J. T., Steinbacher, M., Worsnop, D. R., Baltensperger, U., and Prévôt, A. S. H.: Fourteen months of on-line measurements of the non-refractory submicron aerosol at the*

*Jungfraujoch (3580 m a.s.l.) – chemical composition, origins and organic aerosol sources. Atmos. Chem. Phys., 15, 11373-11398, doi:10.5194/acp-15-11373-2015, 2015.*

*Li, W., and Hopke, P. K.: Initial Size Distributions and Hygroscopicity of Indoor Combustion Aerosol-Particles, Abstr Pap Am Chem S, 206, 53-Envr, 1993.*

*Zhang, Y. P., Sumner, W., and Chen, D. R.: In Vitro Particle Size Distributions in Electronic and Conventional Cigarette Aerosols Suggest Comparable Deposition Patterns, Nicotine Tob Res, 15, 501-508, 10.1093/ntr/nts165, 2013.*

- Tracing air mass origin with the HYSPLIT model has some limitations in a complex terrain as Jungfraujoch. Another powerful tool to determine source emission sensitivities is FLEXPART, which is specifically improuved for the site (e.g. Pandey Deolal et al., 2014). Given the spatial resolution of HYSPLIT I find it hard to thrust interpretations based on such results alone.

*We use HYSPLIT in the context of the local meteorology and to support the aerosol and INP measurements. While we realize HYSPLIT has limitations in complex terrain, it has been widely used in a number of applications and can serve as a reliable (although, qualitative) tool to evaluate air mass sources. We wanted to use a simple tool to assess general air mass sources to corroborate local wind directionality and possible INP sources. One reason we ran trajectories at multiple ending altitudes was to assess any divergence in the results of general air mass transport direction. We also generalize all possible source locations within range of the trajectory pathways and are careful not to point at any one source. We have revised the second half of section 3.1 to insure we do not over interpret the results from HYSPLIT alone and solely use them to corroborate other in situ measurements.*

- To my understanding the collection of cloud rime should only result in impaction of liquid water droplets on the collection plate, which freeze upon collision. In case that the cloud temperature is colder than the activation temperature of INPs, such samples should not contain ice-active particles. However, INP concentrations are very high in rime samples, and orders of magnitudes higher than the aerosol sample INP concentration. Please explain why this is the case.

*For most of the time cloud temperature at the elevation of Jungfraujoch was warmer than -15 °C. Hence, the differences between rime and snow are small at temperatures below that. However, at cloud temperatures above -15 °C we generally found larger concentrations of INP in snow as compared to rime (Figure 7 in the initial manuscript, Figures 5 and 6 in revision). This finding supports your suggestion made in the second sentence of above paragraph. INP concentrations in rime differ from those in aerosol samples by orders of magnitude because INP concentrations in rime are per unit volume of liquid (rime melted for analysis) and those for aerosol are per unit volume of sampled air. Assuming a liquid water content of 0.1 g m$^{-3}$ the INP concentrations in 1 L of rime are already seven orders of magnitude larger than those of aerosol in 1 L of air.*

**Specific remarks:**

Page 1, title: Please specify that you investigate freezing spectra characteristics; readers outside of the INP community might get confused.

*Done.*

Page 1, lines 26 – 28 (and page 8, lines 284 – 285): I believe that with the methods used here, you cannot clearly identify biological, dust, or a mixture between the two. To make such connection in field studies would require a proper assessment of the aerosol particle population and/or of the ice residuals with respect to the biological and dust particle concentration.

*This is based on the body of previous work, which we elaborate upon in the introduction. Hence, why we stated these are "punitive" influences and "possible" sources.*

Page 1, line 30: This statement needs a reference.

*These are widely accepted and understood processes that are quite broad in scope. We provide specific details under the umbrella of this statement in the following sentences, thus, did not think a citation was necessary for conventional wisdom.*

Page 1, lines 33 – 34: The statement on the microphysical impact on precipitation formation in mixed-phase clouds needs a reference.

*Done.*

Page 3, lines 86 – 87: Before you defined "warm temperature INPs" as INPs active > -15°C (page 2, line 72). Please be consistent.

*Fixed. Stopelli et al. (2016) used a different definition of warm temperature INPs, so we omitted their definition to avoid confusion.*

Page 3, lines 97 – 98: Given the different temporal and spatial resolutions of your aerosol, cloud rime and snow samples, how can you explain the exchange of INPs into air, cloud, and precipitation? E.g. a 24-hr aerosol sample might not be dominated by the INP population which caused cloud and precipitation formation. Also, clouds might from far away from the site, as well as precipitation particles before reaching the ground.

*The difference in temporal coverage between snow, rime, and aerosol sampling was due to practical constraints. It forces us to integrate in our interpretation over an air mass including clouds and cloud-free sections. As long as wind direction and planetary boundary layer influence are similar within an air mass, we think this approach can still lead to insights, although a perfect synchrony in sampling all components would of course have been preferable. We now address this issue at the end of section 2.1 (Aerosol, cloud rime and snow collection at Jungfraujoch).*

Page 3, line 111: Please indicate if you refer to volume or mass/standard flow.

*Fixed. Indicated this is volumetric flow.*

Page 4, line 116: What is the collection efficiency of the drums? Is there a size dependency, e.g. an increased loss due to reflection on the stages for larger particles?

*While we have not conducted our own tests for collection efficiency, the DRUM has been intercompared and characterized by previous work (e.g., Cahill et al. (1987); although they used an 8-stage DRUM). Although the study by Bukowiecki et al. (2009) presents testing of collection efficiency tests with a different DRUM model as ours (i.e., they used a 3-stage DRUM), they demonstrate how rotating drum impactors are generally accurate and robust. We have added these references to section 2.1.*

*Bukowiecki, N., Richard, A., Furger, M., Weingartner, E., Aguirre, M., Huthwelker, T., Lienemann, P., Gehrig, R., and Baltensperger, U.: Deposition Uniformity and Particle Size Distribution of Ambient Aerosol Collected with a Rotating Drum Impactor, Aerosol Sci Tech, 43, 891-901, 10.1080/02786820903002431, 2009.*

*Cahill, T. A., Feeney, P. J., and Eldred, R. A.: Size Time Composition Profile of Aerosols Using the Drum Sampler, Nucl Instrum Meth B, 22, 344-348, Doi 10.1016/0168-583x(87)90355-7, 1987.*

Page 5, line 164: You introduce 222Rn as abbreviation for radon, but you do not use it consistently in the text. Or do I miss a major difference between "radon" and "222Rn"?

*We found one instance beyond the first time we introduce as "Radon ($^{222}$Rn)" in the text and the other time in the new Figure 1 caption where we just used "$^{222}$Rn". We have changed this to "radon" for consistency.*

Page 5, line 177: You do not use the abbreviated "TSP" thereafter, therefore I suggest to not introduce it.

*We have removed "TSP".*

Page 5, line 177: I do not understand why an increase in the 48-hour total suspended particle concentration is of interest in this study; given the time resolution of your rime and snow sample is on the order of hours, an overlap between a detected SDE (> 4 hours) and your sample might have occurred.

*This information was based on previous work and the development of the Collaud-Coen et al. (2004) method for detecting SDEs at Jungfraujoch based on the PSI 48-h time resolution PM sampling. The purpose for providing this information is to demonstrate that SDEs do not always lead to substantially high concentrations of PM based on previous work. If 48-h PM value is not increased, then the SDE event was only a weak one or the temporal overlap with the 48-h period was short. We noted days where SDEs were detected, but that does not always translate to a day-long SDE. We have now clarified that this statement was based on previous work.*

Page 5, lines 168 – 169: In previous studies several approaches have been used to assess boundary layer contact of the air mass arriving at Jungfraujoch. To make the distinction between boundary layer influence and free tropospheric conditions more reliable I suggest to use an additional method, as e.g. described by Herrmann et al., 2015.

*We do not think the use of another boundary layer versus free tropospheric conditions method is necessary, as the radon provides in situ observational evidence of when the air arriving at Jungfraujoch was in contact with the boundary layer or not. Additionally, Herrmann et al. (2015) do use $^{222}$Rn at Jungfraujoch and determine that it is "very much in line with the other two approaches" they used during their in-depth analysis. Thus, $^{222}$Rn serves as a viable parameter for determining boundary layer influences at Jungfraujoch.*

Page 6, lines 200 – 201: This is very vague, and not well quantified. A statistical analysis of e.g. mean travel height of the back trajectories for both wind directions would be helpful.

*This contrasts with the reviewer's previous comment on the limitations of HYSPLIT in complex terrain. This is the reason we generalize sources when considering the HYSPLIT results and only use them to provide some corroboration of local meteorology and the aerosol and INP results. We also compare the trajectories qualitatively and relative to each other. A more quantitative assessment would be speculative.*

Page 6, line 220: What is the correlation coefficient? Is the relationship statistically significant?

*The OPS and radon measurements do not necessarily correlate well when evaluating the highest time resolution of data available. What we intended to demonstrate with this statement was that when looking at a day at a time, we see relationships with increases and decreases during this time compared to before or after. We have revised this statement to reflect this intension.*

Page 7, line 221: How do you determine upslope winds out of figure 1b?

*We removed the word "upslope".*

Page 7, lines 236 – 237: Please add another reference for this quite general statement.

*We have added Jaenicke (1980).*

*Jaenicke, R.: Atmospheric aerosols and global climate, Journal of Aerosol Science, 11, 577-588, https://doi.org/10.1016/0021-8502(80)90131-7, 1980.*

Page 7, line 252 – 257: Figures 6c-h are hard to understand in the way they are visualized. Given that you do not observe any relationship between INP concentration and air temperature/wind speed, I suggest to show these results in in the supplementary material.

*We agree. We removed those from the figure (now the remaining panels are shown in the new Figure 6) and simply state that there was not relationship with wind direction and temperature.*

Page 9, line 300: I cannot identify the freezing spectra for this case study. Please highlight this in e.g. figure 7.

*There is not one case that was discussed on page 9, line 300 (it was a continuation of discussing the air mass trajectories for 15 and 16 Feb). Spectral statistics including onset, $T_{10}$, $T_{50}$, and the warm and cold modal temperatures are presented in the new Figure 6, so one can glean how these days were different than the southeasterly or SDE and boundary layer case days.*

Page 9, lines 300 – 302: I find it hard to follow your argumentation since you do not know the source region and transport pathway of the cloud.

*We assume the air mass transport is not only characteristic of the air mass containing the INPs, but also the air mass containing clouds which likely formed during transport to Jungfraujoch. We have noted here, "…assuming the clouds formed along the air mass transport pathways."*

Page 9, lines 331 – 333: In order to strengthen this finding, you could include meteorological maps indicating frontal systems.

*Unfortunately, we are not aware of any achieved meteorological maps containing such information (e.g., in situ radar) that are available free to the public. However, we looked at NCEP/NCAR reanalyses of 600 mb (approximate height of station) geopotential height (https://www.esrl.noaa.gov/psd/data/composites/day/), which corroborate the passing of a cold front from 22 Feb to 24 Feb to the southwest:*

[Figure]

Feb 600 mb geopotential height (m)    23 Feb 600 mb geopotential height (m)    24 Feb 600 mb geopotential height (m)

*We have now indicated in the manuscript that the passage of a cold front was corroborated with NCEP/NCAR reanalysis. We also note in the methods that air mass transport was supported by NCEP/NCAR reanalyses of wind vectors and geopotential height at 600 mb (section 2.3).*

Page 10, lines 354 – 368: This main part of your conclusion is rather open discussion, perspective and recommendation. I do not see relevant conclusions based on the presented results in this section.

*This was intended to serve as broader context, which we did not originally clearly define by the section header. We have changed the name of this section to "Conclusions and broader implications" for clarity.*

Page 18, figure 1: Apparently, the rime and snow collection time was often longer than the actual occurrence of the cloud event, since the grey shading in figure 1a indicated that the cloud events were lasting shorter than the sampling period (figure 1b). If so, please specify in the text.

*We have revised this figure to show daily averages of cloud fraction but have indicated in the methods section (2.1) that sample collection sometimes lasted longer than cloud events.*

Page 18, figure 1: The labels of the y-axis should not only contain the units, but also the property (e.g. "relative humidity (%)")

*Now that the revised Figure 1 has % of RH and cloud cover on the same axis, we did not specify, but rather added this information to the caption.*

[revised manuscript text omitted]

---

## Referee Report (RR1)

**Comments by Gabor Vali on the revised version of "Using freezing spectra ..... Alps" by Jessie M. Creamean, Claudia Mignani, Nicolas Bukowiecki and Franz Conen.**

March 29, 2019

The paper contains data from a field campaign of about one month duration at the Jungfraujoch observatory. Aerosol, rime and snow samples were taken for analyses of ice nucleating particles (INPs).

The main result is a difference between two wind regimes associated with airmass origins that differ in altitude and geography. These are sensible results, in reasonable accord with previous work and with what is known about INP source/activity relationships. Thus, the results here obtained reinforce general understanding but do not yield significant new insights regarding the sources and composition of INPs. What is new in the paper is the use of differential spectra of INP activity. While there are other good measures of INP activity, the use of differential spectra allows a clearer identification of the properties of INPs.

The basic result of the work emerges from the paper with sufficient clarity in spite of the somewhat excessive effort to provide explanations for every detail. Many rationalizations are speculative and qualitative. The connections explored provide some sense of reassurance that no unexpected phenomena need to be invoked, but do not rise to the level of proofs. This is not unusual for material dealing with the multitude of issues here involved: distant aerosol sources, in-cloud processes, precipitation fall-out, complex terrain and experimental difficulties. So, while readers may well find the paper overly burdened with detail, the authors' efforts can be respected.

Regarding the analyses of nucleus spectra, the inclusion of $K(T)$ and $k(T)$ is a clear improvement over the earlier version. As defined by the equations presented, these quantities provide absolute measures of INP concentration per volume of liquid (for rime and snow) and per volume of air (for the aerosol samples). However, a normalization is introduced (line 175) for reasons that are not clear. Perhaps it is done to compare the rime and snow samples with the aerosol samples on the same scale (line 314). That could be accomplished by plotting the two sets of data with different abscissa scales without the loss of absolute values of concentrations. As it is, the data from this paper cannot be compared to other results. Even on a procedural basis there is a problem: maximum values of $k(T)$ are used as normalizing factor for each sample but these maximum values are obtained with very low values (usually 1 or 2) for both $\Delta N$ and $N(T)$ and so the ratio has large uncertainties and arbitrariness. The paper would gain from eliminating this normalization. If the authors insist on retaining it, the maximum values chosen should be presented.

Onset temperature and $T_{10}$ and $T_{50}$ are introduced as metrics for the INP measurements on lines 296-298 and are used in Fig. 6 (b) to (d) and in Section 3.3. This is understandable in view of the frequent use of these metrics in the past. Their use in this paper is regrettable as it goes against the goal stated in the title of the paper and elsewhere that spectra provide the focus of the analyses. The onset

temperature is statistically a weak metric, $T_{10}$ and $T_{50}$ are somewhat better, but both are experiment specific, most significantly because of volume and sample size dependence. They do not constitute independent measures of activity to reinforce what is indicated by $k(T)$ and $K(T)$ and they are easily substituted by more defensible measures. In this paper, the value of $K(T)$ at -15°C, $K_{-15}$, or at some other nearby temperature would be a good way to distinguish between results for different air-flow regimes. To re-state, the results shown in Fig. 6 (b) to (d) are not wrong, but by-pass the stated goal to use spectra measures and are weaker metrics than what the spectra could provide. Thus, the paper looses effectiveness. Comparisons to other works are hindered.

Why $df/dT$ is retained in the analyses is somewhat puzzling. This quantity provides no information different from $k(T)$ for the warm part of temperature range, i.e. for low values of $f(T)$, and is clearly an artifact at the cold end of the data range. As argued in my pervious comment (point #6) $df/dT$ necessarily falls off when $f(T)$ approaches unity and the location of the peak reflects primarily what fraction of the sample drops have already frozen at higher temperatures, not what is significant for the colder temperature region. Should the volume of the drops in the freezing tests been smaller, or should the water samples have been diluted with INP-free water, the $df/dT$ peaks would have shifted to lower temperatures. As it is, the $df/dT$ plots provide some comparison among the samples but are clearly misleading when used as a basis of interpreting the results in terms of INP types.

Phrasing the results in terms of warm mode and cold mode INPs (lines 261-265 and more) is a tempting but imprecise argument. While peaks near -10°C are significant, the cold mode is an artifact. The most common pattern of INP concentration functions is a monotonic increase toward colder temperatures. The monotonic increase can arise for a single material of for mixtures of several types of INPs. In general, the interpretation of signatures in $k(T)$ is not yet well studied. When a peak is found in the differential spectra, the peak it can be assumed to represent either the presence of a specific INP material or a type of site configuration with frequency above the general trend. However, decomposing differential spectra in terms of different contributions will require much further work. The association of peaks near -10°C with biological INP sources, and activity observed below about -17°C with mineral sources should be made with more recognition of its tentative basis than is done in this manuscript. This is specially the case, since no independent analyses of composition, particle surface properties or other potentially relevant parameters are presented in this paper.

Regarding different spectral features, it is remarkable that the slopes of the spectra below about -17°C are very similar for the all snow and rime samples. While near -20°C there is up to factor 100 range in concentration for the snow samples and not much less for the rime samples, the slopes are quite similar for all cases. Aerosol samples show more variability for the lower temperature range but still aren't far from the trend seen in the snow and rime samples. Such features are noteworthy but the underlying causes are not yet clear.

Minor points:

| | |
|---|---|
| line 58 | Seems like the word 'While' is an error |
| line 67 | 'intact' instead of 'in-tact' |
| line 80 | 'intercomparison' doesn't express well the combined use of the data from the three different samples |
| lines 100-101 | There is a substantial difference in how the main goals and accomplishments of the work are stated on lines 80-81 and on these lines. |
| line 109 | 'inherent time' ??? |
| line 117 | 'rime' instead of 'rimed' |
| line 117 | What is known about the collection efficiency as a function of particle size for the intake configuration? |
| lines 145 | The error introduced by variation in drop volume can be ascertained from the equations used for $k(T)$ and $K(T)$ -- it is of direct proportional magnitude. Stating this uncertainty as undetermined is incorrect. |
| line149 | Was the variation in cooling rate due to slowing as the temperature lowered? Was it variable from one experiment to the other? |
| line 162 | ' average for each drop'  ??? |
| lines 188, 195 | Have BLI and SDE been defined? |
| line 331 | 'not all samples contain a warm mode' would be better phrasing |
| line 350 | This is a problem with the use of $df/dT$ for analysis. If all drops froze at temperatures above about -17°C for this sample, the concentration of less active nuclei could have been determined by dilution of the sample with nuclei-free water. |
| line 371 | 'differing' may be better instead of 'variable' |
| Fig. 6a | A legend placed inside the diagram area would be better than the one along the right-side axis. |
| line 373 | At what temperature is this range of concentrations evaluated? |

---

## Referee Report (RR2)

Review of "Using freezing spectra characteristics to identify ice nucleating particle populations during the winter in the alps" by Creamean et al. (2019)

The work by Creamean at el. analyzes the ice nucleating particle (INP) activity within samples of aerosol, rime, and snow collected at the High Altitude Research Station in Jungfraujoch. An interesting and key finding of the work is that samples originating from the southeast of the sampling location contained appreciable INP activity at temperatures above -15 °C. The authors refer to this as a "warm mode" and quantify it with the differential INP function, $k(T)$, as well as the derivative of the frozen fraction with respect to temperature, $df/dT$. In a previous review of this paper, Gabor Vali made a convincing argument that the $df/dT$ metric misrepresents the existence of a bi-modality in the data. I echo this argument and will try to use a synthetic INP case to illustrate why the authors' use of $df/dT$ is problematic.

For my example I will consider a population of $V$ = 1 $\mu L$ droplets containing 1 $\mu$m desert dust particles having the INP activities measured by Niemand et al. (2012). I will also assume 10% of the particles contain Snomax, the INP activity of which can be quantified using the empirical relation from Wex et al. (2015). The left hand panel of Fig. 1 shows the resultant frozen fraction as a function of temperature with 10% of droplets freezing at high temperatures due to the presence of Snomax and the rest of the droplets freezing at colder temperatures following the INP activity of dust. I note that this example has been synthesized to have a warm mode. Now if we analyze $V * k(T)$ and $df/dT$ (the right hand panel of Fig. 1), we can see that at warm temperatures the two metrics are virtually equivalent but diverge at lower temperatures. We can understand why this is the case by investigating the relationship of $f$ to the cumulative INP function, $K(T)$ and $k(T)$ from the original equation by Vali (1971):

$$f = 1 - \exp(-K(T)V) \quad (1)$$

Differentiating $f$ with respect to $T$ we obtain:

$$\frac{df}{dT} = Vk(T)\exp(-K(T)V) \quad (2)$$

Equation 2 implies that when $f \approx 0$:

$$\frac{df}{dT} \approx Vk(T) \quad\quad\quad (3)$$

However, when $f \approx 1$:

$$\frac{df}{dT} \approx 0 \quad\quad\quad (4)$$

(4) must be satisfied for exactly the reason cited by Gabor Vali in his review: "Since fewer and fewer drops are left as the sample drops freeze and $f(T)$ approaches unity at the lowest temperatures in an experiment, the increase in $f(T)$ per temperature interval, $[df(T)/dT]$, can be expected to drop off, just as it seen in these figures."

Based on the argument made previously by Vali and reiterated here, the statement made by the authors on page 8 line 63 is misleading as it is attributing what is analogous to the cold temperature peak of the blue line in the right hand plot of Fig. 1 to a "cold mode". This is simply not the case. However, coincidently, the attribution of the warm peak to the warm mode is valid but only because of the approximation in Eqn. (3).

In their revision, the authors have added the $k(T)$ analysis which is consistent with previous work and is better equipped to analyze the INP activity of the collected samples. $k(T)$ exhibits the warm mode signature the authors are valuably reporting. I therefore urge the authors to remove the $df/dT$ analysis as it would avert any future confusion on how to properly diagnose freezing spectra while preserving the key findings of the paper.

[Figure]

Figure 1. Left: Frozen fraction vs. temperature of a synthetic data set comprised of a population of droplets containing a mixture of dust and biological particles. Right: $df/dT$ (blue) and $k(T)$ of the INP mixture.

References

Niemand, M., Möhler, O., Vogel, B., Vogel, H., Hoose, C., Connolly, P., … Leisner, T. (2012). A Particle-Surface-Area-Based Parameterization of Immersion Freezing on Desert Dust Particles. *Journal of the Atmospheric Sciences*, *69*, 3077–3092. https://doi.org/org/10.1175/JAS-D-11-0249.1

Vali, G. (1971). Quantitative Evaluation of Experimental Results an the Heterogeneous Freezing Nucleation of Supercooled Liquids. *Journal of the Atmospheric Sciences*. https://doi.org/10.1175/1520-0469(1971)028<0402:QEOERA>2.0.CO;2

Wex, H., Augustin-Bauditz, S., Boose, Y., Budke, C., Curtius, J., Diehl, K., … Stratmann, F. (2015). Intercomparing different devices for the investigation of ice nucleating particles using Snomax[®] as test substance. *Atmospheric Chemistry and Physics*, *15*(3), 1463–1485. https://doi.org/10.5194/acp-15-1463-2015

---

## Author Response (AR2)

We would like to sincerely thank Gabor for his feedback on the revision of our manuscript. We have responded to his remaining concerns as discussed in detail below and hope this sufficiently addresses his concerns before publication of the final version. We have highlighted his contributions more than the last version in the acknowledgements.

Regarding the analyses of nucleus spectra, the inclusion of K(T) and k(T) is a clear improvement over the earlier version. As defined by the equations presented, these quantities provide absolute measures of INP concentration per volume of liquid (for rime and snow) and per volume of air (for the aerosol samples). However, a normalization is introduced (line 175) for reasons that are not clear. Perhaps it is done to compare the rime and snow samples with the aerosol samples on the same scale (line 314). That could be accomplished by plotting the two sets of data with different abscissa scales without the loss of absolute values of concentrations. As it is, the data from this paper cannot be compared to other results. Even on a procedural basis there is a problem: maximum values of k(T) are used as normalizing factor for each sample but these maximum values are obtained with very low values (usually 1 or 2) for both  $\Delta N$  and N(T) and so the ratio has large uncertainties and arbitrariness. The paper would gain from eliminating this normalization. If the authors insist on retaining it, the maximum values chosen should be presented.

**Thank you for highlighting this issue. We have removed the normalization from the spectra in all the figures so that absolute K(T) and k(T) can be assessed in comparison with reported values from other studies.**

Onset temperature and  $T_{10}$  and  $T_{50}$  are introduced as metrics for the INP measurements on lines 296-298 and are used in Fig. 6 (b) to (d) and in Section 3.3. This is understandable in view of the frequent use of these metrics in the past. Their use in this paper is regrettable as it goes against the goal stated in the title of the paper and elsewhere that spectra provide the focus of the analyses. The onset temperature is statistically a weak metric,  $T_{10}$  and  $T_{50}$  are somewhat better, but both are experiment specific, most significantly because of volume and sample size dependence. They do not constitute independent measures of activity to reinforce what is indicated by k(T) and K(T) and they are easily substituted by more defensible measures. In this paper, the value of K(T) at -15°C, K-15, or at some other nearby temperature would be a good way to distinguish between results for different air-flow regimes. To re-state, the results shown in Fig. 6 (b) to (d) are not wrong, but by-pass the stated goal to use spectra measures and are weaker metrics than what the spectra could provide. Thus, the paper looses effectiveness. Comparisons to other works are hindered.

We retain the use of onset temperature and  $T_{10}$  and  $T_{50}$  as the other goal was to assess the variability in INP spectra from the different directions from INCAS only, and meant not for quantitative comparison with previous measurements at Jungfraujoch or elsewhere. We have highlighted this goal in the second to last paragraph of the introduction to make this clear – that this is a secondary goal of the reported results.

Why df/dT is retained in the analyses is somewhat puzzling. This quantity provides no information different from k(T) for the warm part of temperature range, i.e. for low values of f(T), and is clearly an artifact at the cold end of the data range. As argued in my pervious

comment (point #6) df/dT necessarily falls off when f(T) approaches unity and the location of the peak reflects primarily what fraction of the sample drops have already frozen at higher temperatures, not what is significant for the colder temperature region. Should the volume of the drops in the freezing tests been smaller, or should the water samples have been diluted with INP-free water, the df/dT peaks would have shifted to lower temperatures. As it is, the df/dT plots provide some comparison among the samples but are clearly misleading when used as a basis of interpreting the results in terms of INP types.

The goal of showing df/dT is it clearly demonstrates modality between the samples from the different air mass directions when comparing samples within the current work. This metric has been used previously by Augustin et al. (2013) as stated on page 6 and has demonstrated utility in that one can compare the modal presence and height in the different temperature regimes. We regard this as a secondary but useful metric, thus, have retained it in the second revision but have made clear its limitations in section 3.2.

Phrasing the results in terms of warm mode and cold mode INPs (lines 261-265 and more) is a tempting but imprecise argument. While peaks near -10°C are significant, the cold mode is an artifact. The most common pattern of INP concentration functions is a monotonic increase toward colder temperatures. The monotonic increase can arise for a single material of for mixtures of several types of INPs. In general, the interpretation of signatures in k(T) is not yet well studied. When a peak is found in the differential spectra, the peak it can be assumed to represent either the presence of a specific INP material or a type of site configuration with frequency above the general trend. However, decomposing differential spectra in terms of different contributions will require much further work. The association of peaks near -10°C with biological INP sources, and activity observed below about -17°C with mineral sources should be made with more recognition of its tentative basis than is done in this manuscript. This is specially the case, since no independent analyses of composition, particle surface properties or other potentially relevant parameters are presented in this paper.

Although we agree the cold mode is an artifact and experiment-specific, it provides insight into the different possible INP sources within the single INP sample population when using only to compare relative to the warm mode. The warm mode in df/dT is at the same temperature as in the differential spectra (as noted in the caption), and thus may be used for inter-sample comparison. Hence, we show the cold mode, but focus our discussion of sample comparison from the different air mass directions on the presence and location of the warm mode. The probability of the biological (warm mode) versus biological + mineral (cold mode) INPs is based on the current body of literature and is described on page 8. To emphasize the tentative basis of biological versus mineral INPs using spectra alone, we added the following sentence to section 3.2: "However, we note the tentative nature of characterizing these modes based on the previous body of literature and that more controlled, quantitative experiments of mixed biological-mineral INPs is needed in the future."

Regarding different spectral features, it is remarkable that the slopes of the spectra below about - 17°C are very similar for the all snow and rime samples. While near -20°C there is up to factor 100 range in concentration for the snow samples and not much less for the rime samples, the slopes are quite similar for all cases. Aerosol samples show more variability for the lower

temperature range but still aren't far from the trend seen in the snow and rime samples. Such features are noteworthy but the underlying causes are not yet clear.

Interesting point! Perhaps something to evaluate in more detailed, control studies in the future.

Minor points:

line 58: Seems like the word 'While' is an error

'While' has been removed.

line 67: 'intact' instead of 'in-tact'

Done.

line 80: 'intercomparison' doesn't express well the combined use of the data from the three different samples

**Changed to 'comparison'.**

lines 100-101: There is a substantial difference in how the main goals and accomplishments of the work are stated on lines 80-81 and on these lines.

See response above. We have added the secondary goal of comparing INP spectra from the different air mass directions.

line 109: 'inherent time' ???

Removed 'inherent'.

line 117: 'rime' instead of 'rimed'

Done.

line 117: What is known about the collection efficiency as a function of particle size for the intake configuration?

While we have not conducted our own tests for collection efficiency, the DRUM has been intercompared and characterized by previous work (e.g., Cahill et al. (1987); although they used an 8-stage DRUM). Although the study by Bukowiecki et al. (2009) presents testing of collection efficiency tests with a different DRUM model as ours (i.e., they used a 3-stage DRUM), they demonstrate how rotating drum impactors are generally accurate and robust. We have added these references to section 2.1.

Bukowiecki, N., Richard, A., Furger, M., Weingartner, E., Aguirre, M., Huthwelker, T., Lienemann, P., Gehrig, R., and Baltensperger, U.: Deposition Uniformity and Particle Size Distribution of Ambient Aerosol Collected with a Rotating Drum Impactor, Aerosol Sci Tech, 43, 891-901, 10.1080/02786820903002431, 2009.

Cahill, T. A., Feeney, P. J., and Eldred, R. A.: Size Time Composition Profile of Aerosols Using the Drum Sampler, Nucl Instrum Meth B, 22, 344-348, Doi 10.1016/0168-583x(87)90355-7, 1987.

lines 145: The error introduced by variation in drop volume can be ascertained from the equations used for k(T) and K(T) -- it is of direct proportional magnitude. Stating this uncertainty as undetermined is incorrect.

Removed 'undetermined'.

line149: Was the variation in cooling rate due to slowing as the temperature lowered? Was it variable from one experiment to the other?

The cold plate unfortunately does not have a uniform controllable cooling rate and typically slowed as the temperature was lowered and as the system reached its lower limit. It was variable from one experiment to the other, but only minorly variable (i.e., within uncertainty).

line 162: ' average for each drop' ???

Removed 'average'.

lines 188, 195: Have BLI and SDE been defined?

Yes, at the end of the introduction.

line 331: 'not all samples contain a warm mode' would be better phrasing

Done.

line 350: This is a problem with the use of df/dT for analysis. If all drops froze at temperatures above about -17°C for this sample, the concentration of less active nuclei could have been determined by dilution of the sample with nuclei-free water.

This is true; however, we did not conduct dilution drop freezing testing thus cannot determine this with certainty. Still, the fact the warm mode was so dominant suggests that the warm temperature INPs were very abundant. For clarity, we did change this sentence to, "...indicating a relatively large contribution of warm temperature INPs as compared to the other samples from the study."

line 371: 'differing' may be better instead of 'variable'

**Done.**

Fig. 6a: A legend placed inside the diagram area would be better than the one along the rightside axis.

Done.

line 373: At what temperature is this range of concentrations evaluated?

Over the entire spectrum (i.e., at all temperatures measured).

[revised manuscript text omitted]

---

## Author Response (AR3)

Dear Ryan,

Thank you for your efforts in insuring we show the correct representation of the INP spectra by obtaining a third review. We appreciate the extra effort and agree that showing show incorrect analyses would be a critical issue. Thus, we have removed any mention of *df/dT* and the "cold mode" in the manuscript. Please let us know if we need to make any further changes.

Kind regards,
Jessie

[revised manuscript text omitted]

---

## Author Response (AR4)

Dear Ryan,

Thanks again for your input. We have added the Boydoun et al. (2017) and Polen et al. (2018) references to sections 3.2 and 2.2, respectively. Please see the tracked changes below.

Kind regards,
Jessie

[revised manuscript text omitted]